# Concurrent photochemical whitening and darkening of ambient brown carbon

Qian Li[1], Dantong Liu[1*], Xiaotong Jiang[1], Ping Tian[2], Yangzhou Wu[1], Siyuan Li[1], Kang Hu[1], Quan Liu[3], Mengyu Huang[2], Ruijie Li[2], Kai Bi[2], Shaofei Kong[4], Deping Ding[2]

[1]Department of Atmospheric Science, School of Earth Science, Zhejiang University, Hangzhou, 310027, China
[2]Beijing Key Laboratory of Cloud, Precipitation and Atmospheric Water Resources, Beijing Meteorological Service, Beijing, 100089, China.
[3]State Key Laboratory of Severe Weather & Key Laboratory of Atmospheric Chemistry of CMA, Chinese Academy of Meteorological Sciences, Beijing, 100081, China
[4]Department of Atmospheric Science, School of Environmental Science, China University of Geosciences, Wuhan, 430074, China

*Correspondence to*: Dantong Liu (dantongliu@zju.edu.cn)

**Abstract**. The light-absorbing organic aerosol (OA), known as brown carbon (BrC), has important radiative impacts, however its sources and evolution after emission remain to be elucidated. In this study, the light absorption at multiple wavelengths, mass spectra of OA and microphysical properties of black carbon (BC) were characterized at a typical sub-urban environment in Beijing. The absorption of BC is constrained by its size distribution and mixing state and the BrC absorption is obtained by subtracting the BC absorption from the total aerosol absorption. Aerosol absorption was further apportioned to BC, primary BrC and secondary BrC by applying the least-correlation between secondary BrC and BC. The multi-linear regression analysis on the factorized OA mass spectra indicated the OA from traffic and biomass burning emission contributed to primary BrC. Importantly, the moderately oxygenated OA (O/C=0.62) was revealed to highly correlate with secondary BrC. These OA had higher nitrogen content, in line with the nitrogen-containing functional groups detected by the Fourier transform infrared spectrometer. The photochemical processes were found to reduce the mass absorption cross section (MAC) of primary OA but enhancement for secondary OA, resulting in the contribution of primary BrC to total absorbance decreased about 20% but enhanced contribution of secondary BrC by 30%, implying the concurrent whitening and darkening of BrC. This provides field evidence that the photochemically produced secondary nitrogen-containing OA can considerably compensate some bleaching effect on the primary BrC, hereby causing radiative impacts.

## 1. Introduction

Atmospheric absorbing organic aerosol (OA), known as brown carbon (BrC), is an important contributor to anthropogenic absorption besides black carbon (BC) (Laskin et al., 2015; Liu et al., 2020), particularly at shorter visible wavelengths (Bahadur et al., 2012). Due to complex compositions of OA, the primary sources and subsequent evolution of BrC in the atmosphere remains to be explicitly understood and causes uncertainties in evaluating the radiative impacts of BrC (Liu et al., 2020).

The chromophores of BrC are mainly aromatic compounds associated with certain functional groups (Liu et al., 2015c). Particularly, compounds containing nitro, nitrated or other forms of nitrogen-containing functional groups are more absorbing (Nakayama et al., 2013; Jacobson, 1999). It is well established that primary OA, especially from biomass burning, contains a large fraction of BrC (Andreae and Crutzen, 1997; Rizzo et al., 2013; Bond, 2001). These primary BrC has a range of absorptivity, which was found to be controlled by burning phases. OA co-emitting with BC (the flaming phase) exhibited a higher absorptivity than OA-dominated smoldering phase (Liu et al., 2021). BrC can experience reactions with atmospheric oxidants after emission. Previous studies (Satish et al., 2017; Satish and Rastogi, 2019; Dasari et al., 2019) found nitrogenous compounds from biomass burning were responsible for BrC over South Asia and the chromophores were photobleached in the afternoon. Numerous field and laboratory studies found the decrease of BrC absorptivity due to photobleaching of chromophores, with lifetime ranging from a few hours (Zhao et al., 2015; Liu et al., 2021) to a few days (Forrister et al., 2015), which may depend on the concentration of ambient hydroxyl radical (Wang et al., 2014), also influenced by relative humidity and particle volatility (Schnitzler et al., 2020). The absorptivity of BrC could be also enhanced due to addition of functional groups by forming conjugated structure with aromatics. This was supported by a number of laboratory studies that BrC absorptivity could be enhanced when forming nitrogen-containing organic compounds, such as the formation of nitro-aromatics when aromatics reacted with $NO_x$ (Nakayama et al., 2013), or produced organic amine after reacting with ammonia (Updyke et al., 2012). The enhancement of BrC absorptivity could occur either through nitration of existing chromophores, or formation of new secondary organic aerosol (SOA) chromophores through gas-phase oxidation.

The above findings mean the enhancement or bleaching of BrC absorptivity via photooxidation will coexist. The time scale between both competing processes will ultimately determine the lifetime of BrC in the atmosphere. However, both processes have been rarely investigated in the field to explicitly determine the BrC components which principally determine the respective enhancement or decrease of its absorptivity, particularly in regions influenced by combined anthropogenic sources. In this study, by measurements using multiple-wavelength absorption and microphysical properties of BC in a sub-urban region, the absorption of BC, primary and secondary BrC was discriminated. In conjunction with source attribution via OA mass spectra, we are able to link the segregated absorption with certain sources and investigate their primary information and subsequent evolution. The competition between photobleaching and secondary formation of BrC was investigated in real world.

## 2. Experimental and instrumentation

## 2.1 Site description and meteorology

The experiment was conducted during springtime at the Beijing Cloud Laboratory and Observational Utilities Deployment Base (117.12°E, 40.14°N), which is located in the northeast suburban area in Beijing (Fig S1a). The site is surrounded by the northwest mountain ridge, without significant local primary anthropogenic emissions (Hu et al., 2021). The 72-h backward trajectories with every 3 hours initializing from the site are analyzed by the HYSPLIT model (Draxier and Hess, 1998) using the 3-hourly 1°×1° meteorological field from the GDAS reanalysis product. The obtained backward trajectories were further clustered to group the similar transport pathways (Makra et al., 2011). The meteorological parameters, including the temperature (T), ambient relative humidity (RH), wind speed (WS) and wind direction (WD) were measured by a monitoring station on the site.

## 2.2 Measurements of BC microphysics and absorption coefficient

In this study, the ambient aerosols were sampled by a large-flow (1.05 m$^3$ min$^{-1}$) air particle sampler (TH-1000C Ⅱ) with a PM$_{2.5}$ impactor (BGI SCC 1.829) and dried by a silica drier before measurement. The single particle soot photometer (SP2, DMT., USA) used continuous laser at $\lambda$=1064 nm to incandesce light-absorbing aerosols (such as BC) for irradiating detectable visible light. The incandescence signal was used to measure the refractory black carbon (rBC) mass. The SP2 incandescence signal was calibrated using the Aquadag standard (Acheson Inc., USA), and a factor of 0.75 was applied to correct for ambient BC (Laborde et al., 2012). The scattering signal was calibrated by monodispersed polystyrene latex spheres (PSL). The BC core diameter ($D_c$) was calculated from the measured BC mass by assuming a BC density of 1.8 g cm$^{-3}$ (Bond and Bergstrom, 2006). The leading edge only (LEO) method was applied to reconstruct the scattering signal of BC, which was used to determine the coated particle diameter ($D_p$) by a Mie-lookup table with the inputs of scattering and incandescence signal of each BC particle (Liu et al., 2014; Taylor et al., 2015). The mass median diameter (MMD) is derived from the $D_c$ distribution, which is determined as below and above MMD the rBC mass concentration is equal (Liu et al., 2019a). The bulk coating thickness ($D_p/D_c$) is calculated as the cubic root of ratio of the total coated BC volume divided by the total volume of rBC. The mass absorption cross section (MAC) (in m$^2$ g$^{-1}$) of each BC particle can be calculated using the measured coated and uncoated BC sizes by applying the Mie core-shell calculation. The absorption coefficient of BC at certain wavelength, $\sigma_{abs,BC}$ ($\lambda$) is determined by multiplying the calculated MAC and rBC mass concentration at each size:

$$\sigma_{abs,BC} (\lambda)=\sum_i MAC(\lambda, D_{p,i}, D_{c,i})m(logD_{c,i})\Delta logD_{c,i} \tag{1}$$

where m ($logD_{c,i}$) denotes the BC mass concentration at each logarithmic bin of $D_c$. The SP2 measurement at $\lambda$=1064nm longer than mostly populated BC size means the derived coatings and subsequent calculation of MAC is relatively independent of particle shape within uncertainty of 21% (Liu et al., 2014; Hu et al., 2021).

The absorption coefficients at wavelengths $\lambda$= 375, 470, 528, 635 and 880 nm were measured by a Micro-Aethalometer (MA200, Aethlabs, San Francisco, CA, USA). Aerosol particles were collected on filter tapes, on which the light attenuation was measured continuously with a time resolution of 30 s. The loading effect of filters was automatically corrected by

measuring attenuation at two different sampling flow rates on two spots in parallel (Drinovec et al., 2015a). Moreover, a multi-scattering correction factor (C-value) of 3.5, 3.2 and 2.4 at the wavelengths 370 nm, 528 nm and 880 nm, respectively were utilized to correct attenuation for the multiple light scattering effect. It was obtained by comparing the absorption coefficient with a photoacoustic soot spectrometer (PASS-3, DMT) (Hu et al., 2021).

**2.3 Attribution of primary and secondary BrC absorption coefficient**

The absorption coefficient of BC at different $\lambda$ is calculated using the measured uncoated core and coated size as mentioned above. The absorption coefficient of total BrC is obtained by subtracting the BC absorption coefficient from the total absorption at certain wavelength, expressed as:

$$\sigma_{abs,\,BrC}(\lambda) = \sigma_{abs,total}(\lambda) - \sigma_{abs,BC}(\lambda) \tag{2}$$

where the absorption coefficient of BC ($\sigma_{abs,BC}$) is obtained from the SP2 measurement, $\sigma_{abs,total}(\lambda)$ is the total light absorption of aerosols measured by the MA200. The absorption coefficient of secondary BrC, the absorption not contributed by primary sources, is obtained by subtracting the absorption of all primary sources from the total absorption (Crilley et al., 2015), expressed as:

$$\sigma_{abs,secBrC}(\lambda) = \sigma_{abs,total}(\lambda) - \sigma_{abs,pri}(\lambda) \tag{3}$$

where $\sigma_{abs,pri}(\lambda)$ is the light absorption from primary sources. Here an assumption is made that light absorption from primary aerosols is all from combustion sources, and these sources necessarily contain BC (Wang et al., 2018a). Therefore, the total absorption from primary sources can be obtained by scaling a factor from the mass concentration of BC, expressed as:

$$\sigma_{abs,pri}(\lambda) = \left(\frac{\sigma_{abs,total}}{[rBC]}\right)_{pri} \bullet [rBC] \tag{4}$$

where $[rBC]$ is the mass concentration of rBC measured by the SP2, $\left(\frac{\sigma_{abs,total}}{[rBC]}\right)_{pri}$ is the scaling factor to derive the absorption of primary combustion sources from $[rBC]$. This factor is obtained using the minimum R-squared (MRS) approach (Wu and Yu, 2016), by adjusting the factor until a minimum correlation between $\sigma_{abs,secBrC}$ and $[rBC]$ is reached because the absorption from secondary sources are least likely to covary with that from primary sources (Wang et al., 2019a). This method has been used in urban and sub-urban environment to obtain the primary BrC associated with combustion sources. Being different from previous studies, an auxiliary characterization of rBC mass measured by the SP2 is used here to avoid the possible interference from absorption measured by the same instrument. There may be different $\left(\frac{\sigma_{abs,total}}{[rBC]}\right)_{pri}$ ratio between traffic and biomass burning sources and this may lead to bias in deriving the subsequent results. We have more carefully investigated the diurnal pattern of hydrocarbon-like OA (HOA) and biomass burning OA (BBOA), and found only a slight morning rush-hour peak for HOA (though bearing considerable variation). A further investigation on the HOA/BBOA ratio found no apparent diurnal pattern (bearing large variation), shown in Fig. S8. The source difference is therefore not considered to have significantly influenced the diurnal pattern of derived parameters. In addition, this method is only valid with sufficient data points thus we

may only obtain a single mean value for the entire experiment, which represents the mean $\left(\frac{\sigma_{abs,total}}{[rBC]}\right)_{pri}$ in this environment
during the experimental period. Previous studies using this method also derived the mean value of $\left(\frac{\sigma_{abs,total}}{[rBC]}\right)_{pri}$ for the urban
environment influenced by multiple sources including traffic, coal combustion and biomass burning (Wang et al., 2019c; Wang
et al., 2020; Gao et al., 2022). The $\left(\frac{\sigma_{abs,total}}{[rBC]}\right)_{pri}$ ratio at $\lambda$=375 nm, 470 nm, 528 nm, 635 nm and 880 nm is calculated to be
20.7, 17.0, 14.4, 11.7 and 5, respectively (Fig. S2), which falls within the reported values from previous studies 11-50 (Zhang
et al., 2020; Wang et al., 2019a). This scenario assumes a relatively consistent absorption relative to BC mass concentration
from sources during experiment. This however may not include some sporadic events when sources with distinct OA or BC
mass fraction may be introduced and alter the single $\left(\frac{\sigma_{abs,total}}{[rBC]}\right)_{pri}$ ratio. The $\sigma_{abs,secBrC}$ therefore represents the overall mean
value during the experimental period but this ratio will vary with seasons and locations. The $\sigma_{abs}$ of primary BrC can then be
calculated as:
$\sigma_{abs,priBrC}(\lambda)= \sigma_{abs,BrC}(\lambda)- \sigma_{abs,secBrC}(\lambda)$ (5)
where $\sigma_{abs,BrC}$ and $\sigma_{abs,secBrC}$ is calculated from Equation (2) and (3), respectively.

## 2.4 Composition measurement

The mass concentration and chemical composition of non-refractory sub-micron PM (NR-PM$_1$) including organic aerosols
(OA), nitrate (NO$_3^-$), sulfate (SO$_4^{2-}$), chloride (Cl$^-$) and ammonium (NH$_4^+$) were determined with a High-Resolution Time-of-
Flight Aerosol Mass Spectrometer (HR-ToF-AMS, Aerodyne Research Inc., USA). The setup, operation, and calibration
procedures of the AMS have been described elsewhere (Canagaratna et al., 2007). During this field observation, the AMS was
operated in V-mode for the quantification of mass concentrations. The composition-dependent collection efficiencies were
applied ((Middlebrook et al., 2012), and the ionization efficiency was calibrated using 300 nm pure ammonium nitrate (Jayne
et al., 2000). Elemental ratios of OA including oxygen-to-carbon (O/C), hydrogen-to-carbon (H/C) and nitrogen-to-carbon
(N/C) were determined to the improved-ambient method (Canagaratna et al., 2015).
Positive Matrix Factorization (PMF) (Paatero and Tapper, 1994) was performed on the inorganic and organic high-resolution
mass spectra to distinguish OA components from different sources (Zhang et al., 2011; Ulbrich et al., 2009; Decarlo et al.,
2010). The mass spectra of the combined matrix for m/z <120 were excluded in PMF analysis. Five OA factors were identified.
The diagnostics of PMF is summarized in Text S1and Fig. S6.

## 2.5 Offline Fourier transform infrared spectrometer (FTIR) analysis

Particulate Matter (PM) samples were collected once a day onto prebaked (600℃,4h) quartz fiber filters (Whatman, QMA,
USA) using a large-flow (1.05 m$^3$ min$^{-1}$) air particle sampler (TH-1000C II). The collected filter samples were stored in the
refrigerator at -20°C before analysis. The infrared spectra of collected samples were measured by a Fourier transform infrared
spectrometer (FTIR, Thermo Scientific, USA) equipped with an iD5 attenuated total reflectance accessory (diamond crystal)
to quantify the chemical functional groups over the wavenumbers range of 550-4000 cm$^{-1}$ with a resolution of 0.5 cm$^{-1}$. The
NO and NO$_2$ symmetric stretch in the FTIR spectra can characterize the functional groups associated with nitrogen-containing
organics (Coury and Dillner, 2008). Fig. S3 shows typical examples of FTIR spectra and the assigned functional groups for
the three pollution levels during experiment. The peak at 1110 cm$^{-1}$ corresponds to the background of the quartz fiber filter
overlapped with some X-H bending vibrations, which is subtracted for the following analysis. The characteristic organic nitrate
spectra appear at wavenumbers 860 cm$^{-1}$ (NO symmetric stretch), 1280 cm$^{-1}$ (NO$_2$ symmetric stretch) and 1630-1640cm$^{-1}$
(NO$_2$ asymmetric stretch) (Bruns et al., 2010). After baseline calibration, The FTIR peaks of 1630cm$^{-1}$ and 860cm$^{-1}$ are
integrated the absorption areas above the baseline. The summed integrated area of -NO and -NO$_2$ are hereby used to indicate
the nitrogen-containing organics. There was no discernable peak of carbonyl group for our infrared spectrum, and the peak of
OH at 2500 cm$^{-1}$ - 3400 cm$^{-1}$ for the carboxylic acid is not discernable neither, thus the influence of ketone and carboxylic acid
may be of less importance for our dataset.
## 3.    Results and Discussion
### 3.1    Source attributed OA
The overview results are shown in Fig. S1. The organics dominated the aerosol compositions for most time, but occasionally
nitrate was the most abundant component (Fig. S1g). Note that the nitrate here may also include components containing in
organics besides ammonium nitrate. Backward trajectories (Fig. S1a-d) showed that the most abundant PM$_1$ concentration was
associated with air masses transported in shorter distance from southern regions (C1), but the longer and faster northerly
transported air mass from cleaner north (C2) could dilute the concentrations.
The resolved OA factors by the PMF analysis are shown in Fig. 1, including the mass spectra, time series and diurnal profiles
of each PMF factor with corresponded external and internal tracers. Three primary OA (POA) were identified as HOA,
cooking-related OA (COA), BBOA, with O/C of 0.31, 0.18 and 0.39 respectively. These POA had considerable fraction of
hydrocarbon fragments (C$_x$H$_y$), indicating their less aged status. The HOA profile was characterized by higher contributions
of aliphatic hydrocarbons and has dominated ion tracers such as $m/z$ 41 (C$_3$H$_5^+$), 43 (C$_3$H$_7^+$), 55 (C$_4$H$_7^+$) and 57 (C$_4$H$_9^+$). The
HOA concentration correlated with BC ($r$=0.62), which emits from traffic emissions. The diurnal variation exhibited strong
morning and afternoon rush-hour peaks of mass concentration. This factor was consistent with the mass spectra of previously
measured HOA from on-road vehicle emissions in urban cities (Zhang et al., 2005; Aiken et al., 2009; Sun et al., 2016; Hu et
al., 2017), which has m/z peaks characteristic of hydrocarbon fragments in series of C$_n$H$_{2n+1}^+$ and C$_n$H$_{2n-1}^+$. The mass spectrum
of HOA shows overall similarity to those of primary OA emitted from gasoline and diesel combustion sources ($r$=0.68) (Elser
et al., 2016).

The OA from cooking sources (COA) is also characterized by prominent hydrocarbon ion series, however, with higher signal at $C_nH_{2n-1}^+$ than $C_nH_{2n+1}^+$. COA had apparent fragments of both $C_4H_9^+$ and $C_3H_3O^+$, and has a higher ratio of $C_3H_3O^+/C_3H_5O^+$ (3.1), $C_4H_7^+/C_4H_9^+$ (2.2) than HOA (0.9–1.1), with cooking-related fragments of $C_5H_8O^+$ (m/z 84), $C_6H_{10}O^+$ (m/z 98) and $C_7H_{12}O^+$ (m/z 112) (Sun et al., 2011b; Mohr et al., 2012). The COA shows overall similar spectral pattern to the reference spectra of COA ($r$=0.92) (Elser et al., 2016). Its minor peak at noon and larger peak in the evening (Fig. 1l) also corresponded with the lunch and dinner time respectively. There was only a minor peak at noon for COA, which may be due to the sub-urban nature of the site where the major aerosols from cooking sources may have been processed and lost the signature near source. The feature of this factor was also observed in sub-urban environment (Huang et al., 2021).

The BBOA factor was identified based on the prominent signals of $m/z$ 60 ($C_2H_4O_2^+$) and 73($C_3H_5O_2^+$), which are known fragments of levoglucosan (Cubison et al., 2011). And BBOA also correlated with potassium ($K^+$, $r = 0.80$), which are indicator of biomass burning (Pachon et al., 2013; Brown et al., 2016). The $m/z$ 60 and 73 together with a unique diurnal variation have been shown to be a robust marker for the presence of aerosols from biomass burning emissions in many urban locations (Sun et al., 2016). The BBOA shows very similar mass spectral patterns to previously reported reference spectra of biomass burning ($r$=0.94) (Elser et al., 2016). The BBOA factor that was identified in spring accounted for 12.8% of the total OA in Beijing, similar to previous reports (Hu et al., 2017). Biomass (Cheng et al., 2013) and solid fuel burning emissions (Sun et al., 2014) have been widely observed to importantly contribute to the primary OA in this region. This off-road combustion source was particularly abundant during wintertime for residential heating activities (Shen et al., 2019; Yang et al., 2018; Liu et al., 2016), while boiler for industry use (mostly using coal as fuel) was in operation throughout the year (Liu et al., 2015b). During the springtime of the experiment, the residential heating activities dropped due to increased ambient temperature thus the BBOA may be mainly contributed by the industry sector.

Two types of oxygenated organic aerosols (OOA) were identified, in moderate (OOA2, O/C=0.62) and high oxidation state (OOA1, O/C=0.95), respectively, which is very similar to the spectra of OOA factors resolved in other cities (Hayes et al., 2013; Ulbrich et al., 2009). The average mass spectrum of OOA2 in this study is characterized by $m/z$ 29 (mainly $CHO^+$), 43 (mainly $C_2H_3O^+$) and $m/z$ 44 ($CO_2^+$), similar to the semi-volatile OOA spectrum identified in other locations (Sun et al., 2011a; Zhou et al., 2016). On average, OOA2 accounts for 42% and 18% of $C_xH_yO^+$ and $C_xH_yO_2^+$ ions, respectively (Fig. 1b). These results clearly indicate that OOA2 was primarily composed of less oxygenated, possibly freshly oxidized organics. Notably, OOA2 had a substantially higher N/C than other factors (N/C=0.037), and had highest correlation with nitrate ($r$=0.77) and with $C_xH_yN_z$ and $C_xH_yN_zO_p$ fragments ($r$=0.83). This factor therefore tends to largely result from nitrogen-containing OA and its elevation at night may be also associated with dark oxidation by nitrate radical.

The mass spectrum of OOA1, which was characterized by a dominant peak at $m/z$ 44 (mainly $CO_2^+$), a highest O/C (0.95). On average, OOA1 contributes 51% of the $C_xH_yO^+$ signal and 23% of the $C_xH_yO_2^+$ signal (Fig. 1a). OOA1 showed particularly high correlation with sulfate ($r$=0.40) because of their similar volatilities (Huffman et al., 2009; Jimenez et al., 2009). The slight enhancement at noon for OOA1 (also for OOA2) soon after morning rush-hour indicated the likely rapid formation of SOA through photooxidation. This significantly higher mean OOA2 than median value in the diurnal pattern indicated that

this OA type was largely associated with pollution events. Both OOA1 and OOA2 showed nighttime peak maybe due to
reduced boundary layer.

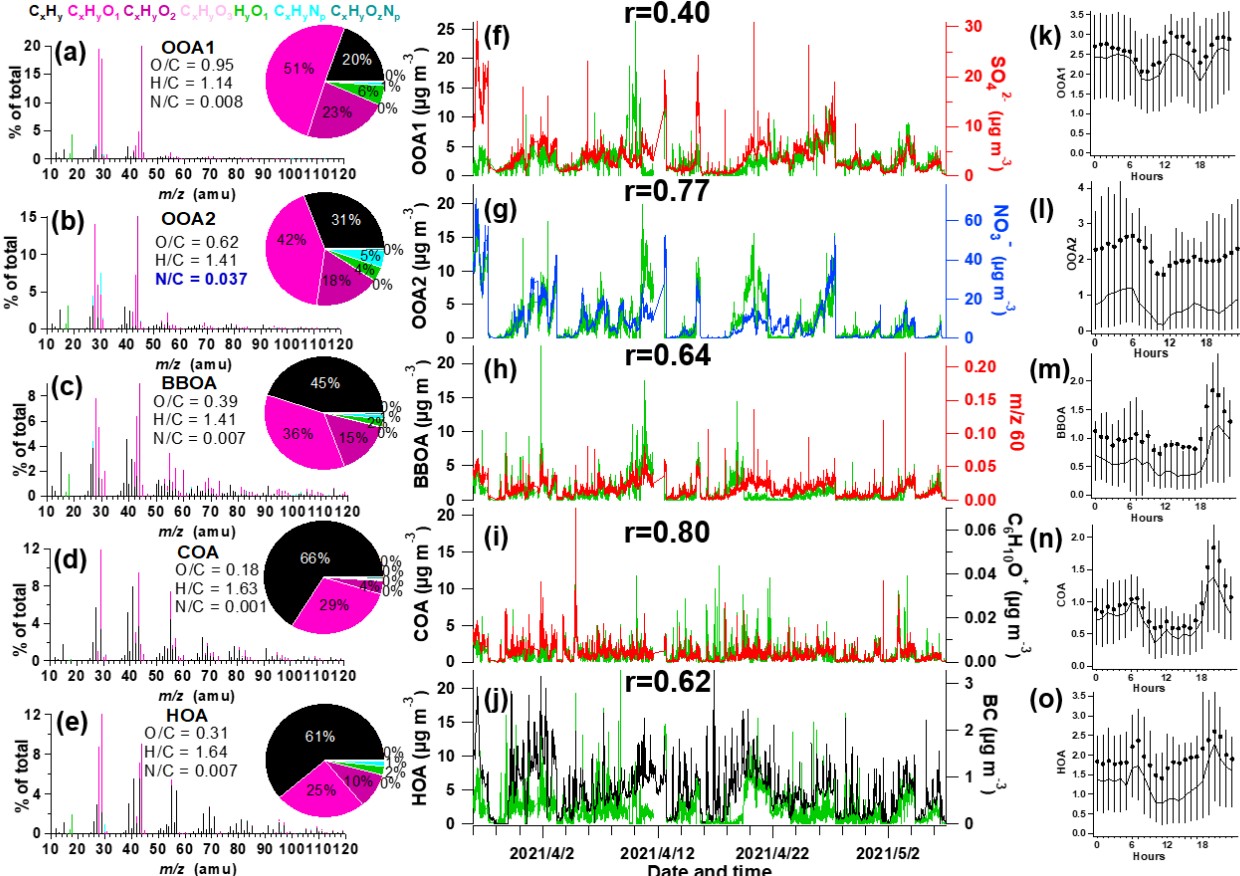

**Figure 1. Information of source-apportioned organic aerosols by the PMF analysis. Mass spectra of (a) oxygenated OA1 (OOA1), (b)**
**oxygenated OA2 (OOA2), (c) biomass burning OA (BBOA), (d) cooking-related OA (COA), (e). hydrocarbon-like OA (HOA), (f-j)**
**Temporal variations of each PMF factor and the corresponding marker species. (k-o) Diurnal profiles of each factor. The lines, dots**
**and whiskers denote the median, mean and the 25th/75th percentiles at each hour respectively.**
**3.2 Segregated aerosol absorption**
Fig. 2 shows the time series of BC properties, including the BC mass concentration, $D_p/D_c$, $D_c$, MAC and light absorption
coefficient of BC (section 2.2). The MMD of BC core varied between 93 – 274 nm which may correspond to the source-
specific information (Liu et al., 2019b) or coagulation process during ageing. The coating of BC (indicated by $D_p/D_c$) showed
sporadic enhancement which was closely associated with enhanced PM concentration (Fig. 2a). This was consistent with
previous studies that high coatings of BC occurred during heavier pollution due to the enhanced secondary formation of
condensable materials to particle phase (Ding et al., 2019; Zhang et al., 2018). This clearly indicates the variation of mixing
state of BC and this will potentially influence its MAC and absorption Ångström exponent (AAE) (Liu et al., 2015a). It will
introduce considerable uncertainties to use constant MAC or AAE to derive the absorption coefficient of BC at multiple
wavelengths. The MAC estimated using the measured BC core size and coatings (Fig. 2c) is thus used to derive the $\sigma_{abs,BC}$
(section 2.2, shown in Fig. 2d). The $\sigma_{abs,BC}$ was 9.1±7.3 Mm$^{-1}$ during experimental period. MAC of BC at $\lambda$=375nm showed to
be at 8.4 -16.6 m$^2$ g$^{-1}$ with enhanced absorption when high coatings, which was consistent with previous studies which reported
MAC$_{BC}$ of 8-10 m$^2$ g$^{-1}$, and higher value of 9.7 -17.2 m$^2$ g$^{-1}$ under polluted condition (Ding et al., 2019; Hu et al., 2021). The
uncertainty of $\left(\frac{\sigma_{abs,total}}{[rBC]}\right)_{pri}$ is 4% for the data points over 1.5 according to (Wang et al., 2019a). The measurement of rBC
mass from the SP2 had uncertainty of 20% (Schwarz et al., 2008), with relative coating thickness having uncertainty of 23%
(Taylor et al., 2015), hereby resulting in a uncertainty of 27% for calculated MAC$_{BC}$. The above results in uncertainties of 31%
and 20% for $\sigma_{abs,BC}$ and $\sigma_{abs,pri}$, respectively. The absorption measurement by MA200 had uncertainty of 25% (Drinovec et al.,
2015b; Duesing et al., 2019). All these uncertainties propagates the uncertainties of $\sigma_{abs,BrC}$, $\sigma_{abs,priBrC}$ and $\sigma_{abs,secBrC}$ as 40%, 37%
and 32% respectively. These are summarized in Table S1.

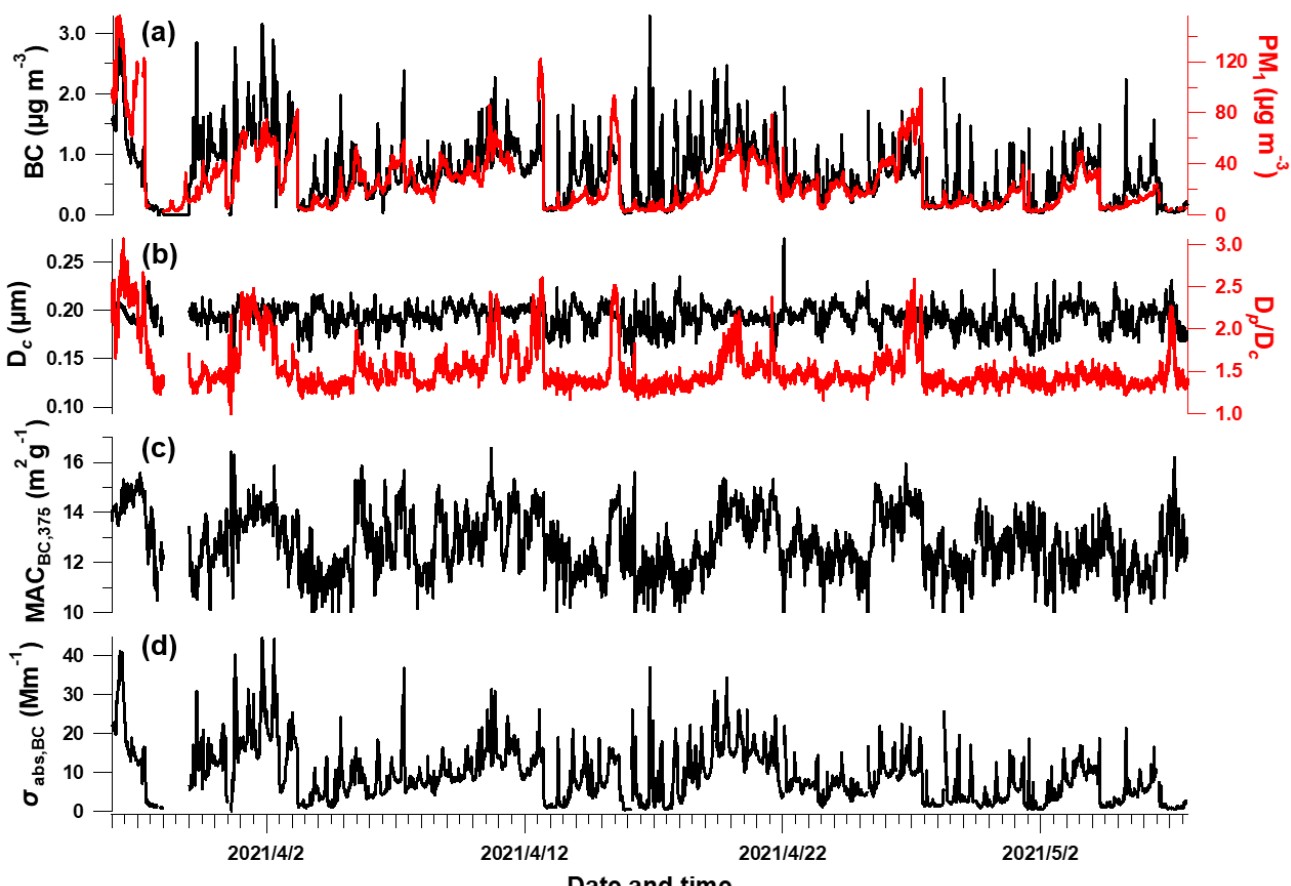

**Figure 2. Temporal evolution of BC-related properties. (a) rBC and PM$_1$ mass concentration, (b) BC core diameter and bulk coating**
**thickness (D$_p$/D$_c$), (c) calculated mass absorption cross section (MAC) at $\lambda$=375nm, (d) absorption coefficient of BC.**
Using the method above, the total ($\sigma_{abs,total}$) and attributed absorption of BC ($\sigma_{abs,BC}$), primary ($\sigma_{abs,priBrC}$) and secondary BrC
($\sigma_{abs,secBrC}$) at $\lambda$=375nm are shown in Fig. 3a-c. In Fig. 3b, the brown and green shades above the adjacent tracer indicate the
absorption coefficient of primary and secondary BrC, respectively. Fig. 3c shows that the absorption coefficient of primary
BrC was higher than secondary BrC for most time, but for certain periods they were equivalent or secondary BrC occasionally
exceeds primary BrC. The mean contribution of absorption coefficient for BC, primary BrC and secondary BrC is 51%, 27%
and 22% in this study. The tracers associated with nitrogen-containing organics, such as OOA2 (with highest N/C), $C_xH_yN_z$
and $C_xH_yN_zO_p$ fragments, and the FTIR measured -NO + -NO₂, are also shown in Fig. 3d-e.

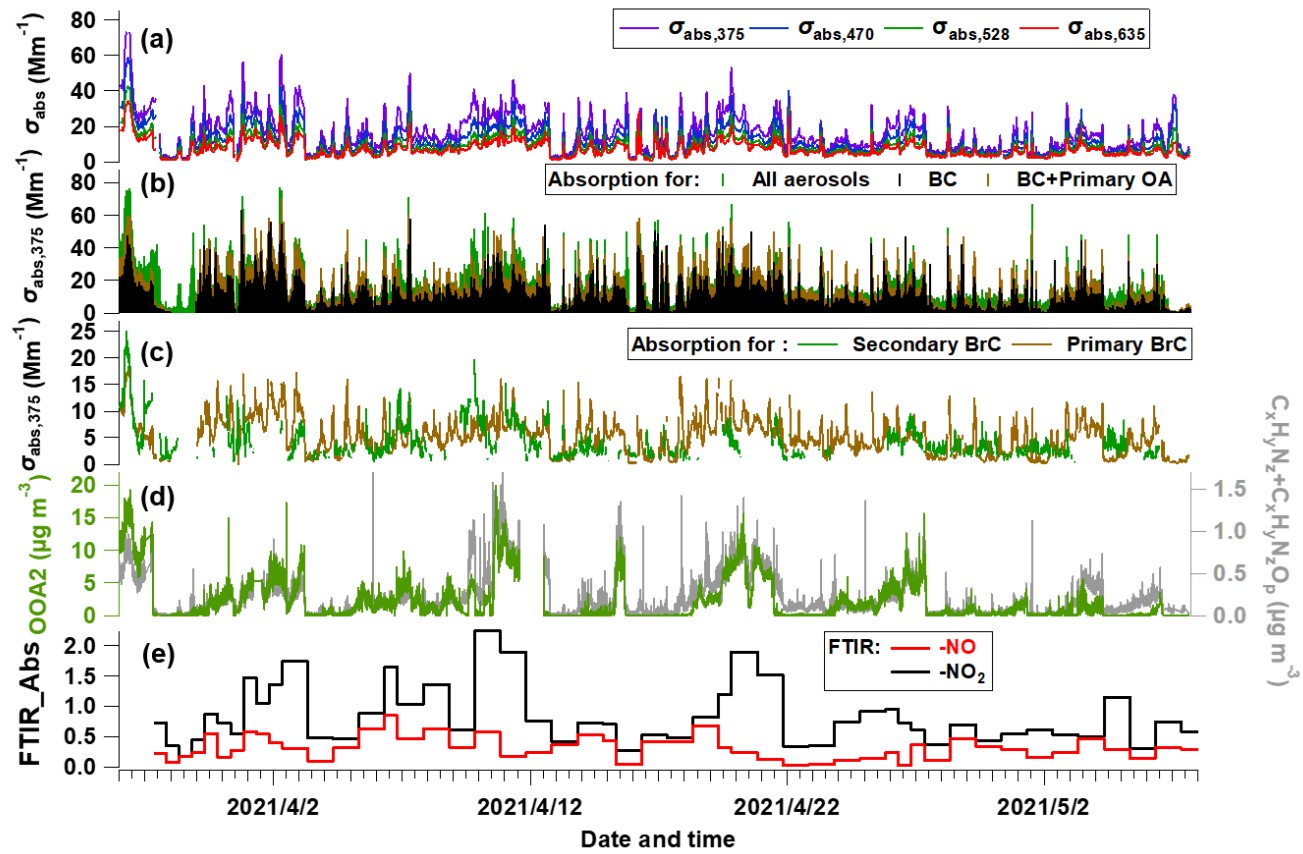

**Figure 3. Temporal evolution of segregated absorbing properties. (a) Absorbing coefficients ($\sigma_{abs}$) at multiple wavelengths measured**
**by the aethalometer, (b) $\sigma_{abs}$ at $\lambda$=375nm ($\sigma_{abs,375}$) for all aerosols, primary OA and BC, (c) $\sigma_{abs,375}$ for primary BrC and secondary**
**BrC. (d) mass concentration of OOA2 and the $C_xH_yN_z$ and $C_xH_yN_zO_p$ fragments measured by the AMS. (e) FTIR-measured**
**absorption of -NO and -NO₂ bonds.**
**3.3  Source attribution of BrC absorption**
A multiple linear regression (MLR) analysis is performed to apportion the absorption coefficient of BrC with the PMF
attributed OA factors, expressed as:
$\sigma_{abs,BrC}=a_0+a_1\cdot[OOA1]+a_2\cdot[OOA2]+a_3\cdot[BBOA]+a_4\cdot[COA]+a_5\cdot[HOA]$ (6)
where $a_1$ to $a_5$ represents the regression coefficients for each factor. These coefficients can be associated with the absorptivity
of each factor, i.e., a larger coefficient implies a higher MAC for the source associated with that OA factor (Kasthuriarachchi
et al., 2020; Wang et al., 2021). The BBOA was found to have the highest MAC at 2.59 m$^2$ g$^{-1}$, consistent with previous studies
which also found significantly higher absorption for biomass burning source (Qin et al., 2018; Wang et al., 2019b; Zhang et
al., 2022). The other POA factors generally have a higher MAC than SOA (the MAC of HOA and COA are is 1.70 m$^2$ g$^{-1}$ and
1.30 m$^2$ g$^{-1}$, respectively). Particularly, the OOA2 has a relatively high MAC of 1.22 m$^2$ g$^{-1}$, which is likely to result from the
production of secondary BrC as discussed below. The contribution of each source-specific OA factor to $\sigma_{abs,BrC}$ can also be
obtained. This analysis is performed for the total BrC, primary and secondary BrC respectively. The results are shown in Table
1. MLR on the total BrC shows relatively higher correction ($r>0.4$) with the factors of HOA, BBOA and OOA2, suggesting
the potential importance of the primary biomass burning and traffic source along with OOA2 in governing absorption of BrC.
MLR analysis on the primary BrC distinguishes its substantial correlation with BBOA ($r=0.40$) and HOA ($r=0.46$), while MLR
on the secondary BrC has a high correlation with OOA2 only ($r=0.44$). The MLR analysis links the apportioned absorption of
physical properties with source-attributed chemical compositions, therefore validating and identifying the sources of primary
and secondary BrC.
**Table 1. Results of the multilinear regression analysis (MLR) between $\sigma_{abs,375}$ and the five PMF-resolved OA factors, with $\sigma_{abs,375}$ of**
**total BrC, primary and secondary BrC as dependent, respectively. All regression coefficients have passed the significance test with**
**p<0.01. Partial correlations above 0.4 are marked in bold. Since negative values appear when the COA participates, which is thus**
**not included in the final regression but the values using COA factor are shown in brackets.**

| Dependent | $\sigma_{abs,BrC}$ | | $\sigma_{abs,pri\,BrC}$ | | $\sigma_{abs,sec\,BrC}$ | |
|---|---|---|---|---|---|---|
| Model | Regression coefficient | Partial correlation | Regression coefficient | Partial correlation | Regression coefficient | Partial correlation |
| Constant | 2.26 | | 1.67 | | 1.47 (1.52) | |
| OOA1 | 0.57 | 0.23 | 0.04 | 0.02 | 0.46(0.46) | 0.24 (0.24) |
| OOA2 | 1.22 | **0.53** | 0.37 | 0.25 | 0.74 (0.74) | **0.44 (0.44)** |
| BBOA | 2.59 | **0.46** | 1.22 | **0.40** | 1.14 (1.18) | 0.29 (0.29) |
| COA | 1.30 | 0.22 | 1.45 | 0.36 | / (-0.25) | / (-0.05) |
| HOA | 1.70 | **0.47** | 1.17 | **0.46** | 0.49 (0.52) | 0.20 (0.21) |
| R$^2$ | 0.77 | | 0.63 | | 0.55 (0.55) | |

Importantly, an oxygenated secondary OA factor (OOA2) is identified to significantly contribute to the secondary BrC. This
OOA has a moderate O/C (0.62) and a highest N/C of 0.037 among all factors. The high N/C means this factor contains the
most abundant nitrogen-containing fragments, implied as its high correlation with the $C_xH_yN_z$ and $C_xH_yN_zO_p$ fragments ($r=0.83$,
Fig. 3d) and with the FTIR absorption for -NO$_2$ and -NO bonds ($r=0.69$, Fig. S4). The -NO bond is mostly related to the
organic nitrates (RONO$_2$), and -NO$_2$ peak could result from both organic nitrates and nitro-organics (Bruns et al., 2010). There
is no discernable peak for organic amines. These all consistently imply that the OOA2 factor contained substantial fraction of
nitrogen-containing organics, and these compounds have contributed to the absorption of secondary BrC.

### 3.4 Simultaneous whitening and darkening process of BrC

The relative contribution and diurnal variation of primary and secondary BrC measured by MA200 at 470, 528 and 635nm
wavelengths are similar to those at 375nm wavelengths, but with decreased fraction of BrC absorption with increased
wavelength. The mean AAE of total BrC, primary BrC and secondary BrC is obtained by power fitting on the mean absorption
coefficient during the experiment (Fig.S7), which is 6.16, 5.69 and 6.40 respectively. This is consistent with other studies that
SOA usually had a higher AAE than POA (Gilardoni et al., 2016; Jiang et al., 2022).Due to the high contribution of BC to total
absorption (>50% even at shortest wavelength), the spectral dependence of absorption in bulk has not shown apparent diurnal
variation. The diurnal variation of $\sigma_{abs,375}$ for BC and primary BrC and their fractions showed consistent morning rush-hour
peaks at 6:00-8:00 and the night-time enhancement due to reduced boundary layer (Fig. 4a-b). This was in line with the morning
peak of HOA and night peak of BBOA. The traffic source in this region, in particular the diesel vehicles, was reported to emit
considerable OA with certain chromophores, such as aromatics (Yao et al., 2015) and heterocyclic organic compounds (Gentner
et al., 2017; Schuetzle, 1983). In the morning rush-hour, BC and primary BrC accounted for 51±4% and 29±4% in the total
$\sigma_{abs,375}$ respectively, with the remaining 20±2% classified as secondary BrC. The morning peak coinciding with the primary
BrC may result from the rapid formation of BrC from sources when emitted gases condensed and formed aerosols. These may
lead to high cooccurrence between primary and secondary BrC. Previous studies in urban environment also observed
concurrent peaks of primary and secondary BrC, which usually occurred at morning rush hour (Zhang et al., 2020).
Furthermore, the assumption of the method used to apportion primary and secondary BrC will cause some error in the
distinction of absorption coefficient, it is possible that some of the primary sources are being attributed to secondary sources
and vice versa. This maybe a possible reason for the simultaneous peak observed for primary and secondary BrC during
morning rush hour. The night had contributions from BC and primary BrC at 50±2% and 30±3% respectively, with 20±3% as
secondary BrC. Fig. 4b showed the decrease of primary BrC absorption tended to be more rapid than the HOA and BBOA
mass (even a slight increase for HOA Fig. 1m and Fig. 1o) in the midday, leading to decreased absorption coefficient per unit
mass of primary BrC (shade in Fig. 4b), which indicates the decrease of BrC absorptivity likely due to photochemistry. This
may involve the OH radical reaction with existing chromophores in aerosol phase (Schnitzler et al., 2020) or by enhanced
evaporation of aerosols to gas phase (Palm et al., 2020) leading to further decrease of BrC absorptivity during midday. In
addition to photobleaching, it possible that some primary species transformed into less absorbing secondary BrC species.
During this period, the type of HOA or BBOA that contribute to absorption may also have a lower absorptivity. In this context,
a recent chamber study reported that the primary BrC from biomass burning plumes could be bleached to half of the initial
absorptivity in 2-3 hours (Liu et al., 2021). The reaction of BrC with OH radical has been widely recognized as the main
pathway for the loss of primary BrC absorptivity (Liu et al., 2020), and was parameterized as an exponential decrease with
time at certain OH radical concentration in global scale (Wang et al., 2018b).
Besides the morning rush-hour peak, there was an early afternoon peak for the absorption coefficient of secondary BrC,
prevailing the dilution effect of daytime boundary layer (Fig. 4c-S5). The night and morning peak of OOA2 and the morning
peak of $\sigma_{abs,secBrC}$ may result from primarily emitted moderately oxygenated OA, which was reported from some diesel sources
(Dewitt et al., 2015; Gentner et al., 2012). The fraction of secondary BrC thus had a pronounced early afternoon peak soon
after the peak solar radiation (Fig. 4f) and a peak after midnight soon after the nighttime peak of primary BrC (Fig. 4e). Fig.
4b showed that the MAC of POA decreased after the morning peak. The MAC of SOA showed an afternoon peak (Fig. 4c),
indicating the enhancement of absorption efficiency of secondary BrC, which occurred in a few hours after the peak solar
radiation. This means the photochemistry caused the absorptivity of POA decreased but the absorptivity of SOA increased. Fig
4e-f shows the photochemical processes led to an enhanced contribution of secondary BrC to the total absorption by 30% from
the morning rush-hour to midday, but during the same time reduced the contribution of primary BrC to the total absorption
about 20%. Though the other process such as aqueous reactions at nighttime may also contribute the change of MAC for BrC,
the apparent change in the daytime was indeed observed in this study, and the absorption of aerosols plays a more important
role on the radiative impacts in the daytime when intensive solar radiation. This shift of peaking time from primary to secondary
BrC demonstrates the likely process of SOA formation from gases, and these SOA compounds containing nitrogen (i.e., the
OOA2) considerably contributed to the light absorption. This ageing or oxidation likely occurred through photooxidation
during early afternoon and aqueous processes (high RH conditions prevail during nighttime) during nighttime. The oxidized
volatile organic compounds (VOCs) with nitrogen chemistry involved could condense to produce additional mass in particle
phase (Ehn et al., 2014; Finewax et al., 2018). Due to the high $NO_x$ emission, photooxidation of traffic VOCs may have largely
involved nitrogen chemistry. Previous studies found the $NO_x$-involved SOA could produce considerable chromophores (Lin et
al., 2015; Siemens et al., 2022), such as the traffic VOCs may produce SOA in a time scale of hours, containing nitro-aromatics
(Wang et al., 2019d; Keyte et al., 2016). The daytime formation of organic nitrate may follow the gas-phase photooxidation
mechanism, in which the excess NO could add to the peroxy radical to produce organic nitrate (Liebmann et al., 2019). The
nighttime chemistry involving $NO_3$ radical through the oxidation of $NO_2$ by $O_3$, contributed to the important formation of
organic nitrate by initializing the production of nitrooxy peroxy radicals (Ng et al., 2008; Rollins et al., 2012). Laboratory
studies (Nakayama et al., 2013; Liu et al., 2015c) also widely observed the rapid production of nitrogen-containing OA
involving $NO_x$ chemistry could contribute to light absorption of aerosols.
Overall, by apportioning the absorption of primary and secondary BrC, the BBOA was found to have the highest MAC and
the other POA factors generally have a higher MAC than SOA, the OOA2 has a relatively high MAC which is likely to result
from the production of secondary BrC. We found the photochemical processes decreased the MAC of POA but increased the
MAC of SOA, resulting in an enhanced contribution of secondary BrC to total absorbance by 30% but reduced contribution
of primary BrC about 20% in the semi-urban environment. This revealed that the whitening and darkening of BrC occurred
simultaneously, and the secondary BrC produced by photooxidation may compensate some bleaching effect of primary BrC.
The dominance of both competing processes may depend on the timescale and altitude in the atmosphere. For example, the
enhanced BrC fraction observed above the planetary boundary layer may be explained by the enhanced secondary BrC (Tian
et al., 2020), while further ageing may bleach the produced chromophores of these SOA.

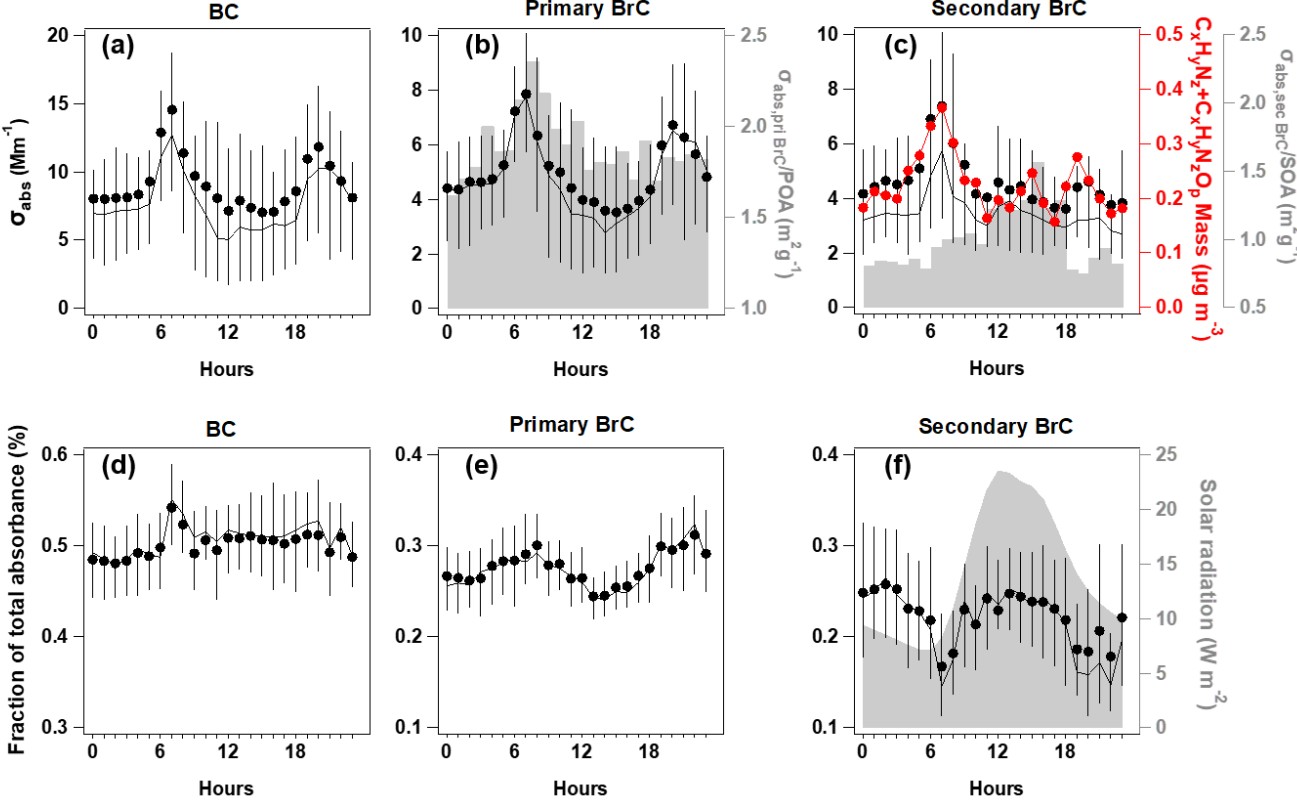

**Figure 4. Diurnal variations of absorption coefficient at λ=375nm ($\sigma_{abs,375}$) for BC (a), primary BrC and absorption efficiency of**
**primary BrC ($\sigma_{abs,pri\ BrC}$)/POA is shown in shade (b), secondary BrC and absorption efficiency of secondary BrC ($\sigma_{abs,sec\ BrC}$)/SOA is**
**shown in shade, along with the $C_xH_yN_z$ and $C_xH_yN_zO_p$ fragments (c); the respective fraction in total for the segregated $\sigma_{abs,375}$ (d-f),**
**with direct radiation shown in shade. In each plot, the lines, dots and whiskers denote the median, mean and the 25th/75th percentiles**
**at each hour respectively.**

## 4. Conclusion

This study apportioned the shortwave absorption of BC, primary and secondary BrC, through concurrent measurements of BC
microphysical properties and OA mass spectra. The apportioned primary BrC absorption was linked with traffic and biomass
burning emissions, while secondary BrC was found to be associated with an oxygenated secondary OA factor with higher
nitrogen content. The enhancement of secondary BrC and decease of primary BrC simultaneously occurred via daytime
photooxidation. The results emphasize the importance of nitrogen-containing OA in contributing to BrC. These OA could
primarily emit as aerosol phase, or in gas phase which requires further oxidation to be in aerosol phase to serve as BrC. The
$NO_x$-involved chemistry is prone to add nitrogen element to the existing OA and enhance the absorptivity of chromophores.
The anthropogenic $NO_x$ emission could be therefore an important source in producing shortwave absorbing components in the
atmosphere, which may offset some of the conventionally-thought photobleaching of BrC by photochemistry. The production
of secondary BrC should be considered when assessing the environment and climate impacts of light-absorbing aerosols.

## Acknowledgments

This research was supported by the National Natural Science Foundation of China (Grant No. 42175116 and 41875167),
National Key R&D Program of China (2019YFC0214703).

## Author contribution

D.L., X.J. and Qian L. prepared and designed the observation. D.L., Qian L., X.J and P.T. initiated the field campaign and
conducted the measurements. Qian L., D.L. P.T., Y.W., S.L. and K.H. contributed to the data analysis. Quan L., H.M., L.R.,
B.K., D.D. and S.K. provided technical support and assistance. Qian L. and D.L. wrote the manuscript. All authors read and
approved the final manuscript.

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
