# Peer review of "Concurrent photochemical whitening and darkening of ambient"

_Atmospheric Chemistry and Physics, 2022_

## Referee Comment (RC1)

**General Comments**: BrC aerosols are short-lived climate forcers and contribute substantially to anthropogenic radiative forcing. Their sources and evolution pathways need to be elucidated. This manuscript titled "Concurrent photochemical whitening and darkening of ambient brown carbon" explores these research questions using diurnal measurements of microphysical (SP2), light absorption (Aethalometer) and chemical characteristics (HR-Tof-AMS and FTIR) at a sub-urban site in Beijing. The manuscript assessed diurnal variation of AMS based PMS source factors, apportioned absorption coefficient at 375 nm for BC, primary BrC and secondary BrC and multiple linear regression between absorption and PMF factors. Overall, the study has some interesting findings about bleaching and darkening of BrC during night time, daytime (photochemical oxidation) and role of nitration in governing these BrC behaviours. However, the manuscript has many shortcomings in its current version. It needs through language editing and clarifications at many places throughout the manuscript. The study has relevance to the atmospheric research community and can be accepted for publication in the journal after major revision. The detailed comments are given below:

**Major comments**

**1. Introduction** Motivation is weak and objectives of study are not clear? Many studies (some of them carried out in Asia are given below) have assessed diurnal profile of BrC absorption and role of nitrogen in governing them. You can cite these paper and please explicitly state how your study is different from these.

*R Satish, N Rastogi On the use of brown carbon spectra as a tool to understand their broader composition and characteristics: a case study from crop-residue burning samples. - ACS omega, 2019. https://doi.org/10.1021/acsomega.8b02637*

*R Satish, P Shamjad, N Thamban, S Tripathi, N Rastogi Temporal characteristics of brown carbon over the central Indo-Gangetic Plain. - Environmental science & technology, 2017. https://doi.org/10.1021/acs.est.7b00734*

2. Section 2.3 Citation for equation 3 and 4 missing? Many previous studies have used primary species, e.g., EC, K+ etc. for quantifying primary and secondary OC. The author can cite those papers. Moreover, relevance or applicability of assumptions taken in eq. 3 and 4 for the site are missing. Please add a brief discussion about all these aspects.

Further, BrC and BC emissions from different sources are very different. For e.g., vehicular emissions are highly rich in BC, but not in BrC. For biomass burning, its vice-versa. How these scenarios will impact the [σabs/[rBC]pri ratio and σabs-SOA estimation. The cluster analysis (Fig S1) and AMS results indicate that scenario is likely (Fig. 1) at the sampling site. How this will impact the overall findings of this study.

3. Line 104-105 How did the authors account for the effect of coating thickness while calculating [σabs/[rBC]pri at different wavelengths?

4. Section 3.1. (Lines 162-167): The authors reported that "Both OOA1 and OOA2 showed nighttime peak due to the dark oxidation chemistry under high relative humidity." But this may or may not be true as boundary layer height is also lower during night compared to daytime. Moreover, nitrate radicals govern the dark oxidation chemistry. Thus, nitration of organics

during nighttime is a possibility, but that was not the case for OOA1 (N/C remain unchanged). Therefore, how can you attribute increase in OOA1 during night to dark oxidation chemistry? Please elaborate.

5. Line 190-191. How did you come with these numbers? Please mention it probably in methodology

If these are based on σabs values, then don't use words such as "mean contribution of absorption for BC, primary BrC and secondary BrC" as σabs values were not weighted with corresponding solar flux values. Instead, you can use words such as "mean contribution of absorption coefficient for BC, primary BrC and secondary BrC". Please keep this point in mind throughout the manuscript.

6. Discussion about some figures is missing in text, e,g, Fig. 4a

7. Line 229-230 and elsewhere: It is mentioned that "photobleaching process causing the decreased absorption efficiency per unit mass for primary BrC." But authors have not provided any discussion about MAC or absorptivity of BrC throughout the manuscript. It is absorption coefficient they are talking about. Please careful look into it.

8. Line 250 "Overall, by apportioning the absorption of primary and secondary BrC, we found the photooxidation led to an enhanced 251 contribution of secondary BrC by 30% but reduced contribution of primary BrC about 20% in the semi-urban environment." How did you come up with these numbers, discuss in either methodology or supplementary.

**Minor Comments**

1. Line 27. This sentence looks confusing. I will suggest to replace the word "shortwave absorption" to "anthropogenic absorption" or "anthropogenic radiative forcing"

2. Line 37: Replace "A range" to "Numerous"

3. Line 39-40: "which may depend on the concentration of ambient hydroxyl radical (Wang et 40 al., 2014)". This is only partially correct. Recently, some studies have reported substantial role of atmospheric condition (RH and temperature, viscosity etc.) on photochemical oxidation. For example.

*Emerging investigator series: heterogeneous OH oxidation of primary brown carbon aerosol: effects of relative humidity and volatility, 10.1039/D0EM00311E, Environ. Sci.: Processes Impacts, 2020, 22, 2162-2171*

Please modify the sentence and cite them properly

4. Line 36-40: The references cited didn't use absorptivity for half-life calculation. All these studies used BrC absorbance to indicate bleaching and BrC lifetime calculation. Please modify your sentence accordingly

5. Line 43-45: revise it to something like "The enhancement of BrC absorptivity could occur either through nitration of exiting chromophores, or formation of new secondary organic aerosol (SOA) chromophores through gas-phase oxidation"

6. Line 48 "rule out" doesn't suit here. Replace it

7. Line 64 Grammatical error, should be "ambient aerosols were"

8. Line 64-65 sentence not clear, revise it.

9. Line 65 should be "…..refractory black carbon (rBC) mass."

10. Line 66-68 Add a little bit more detail in this context.

11. Line 71-72 "The mass median diameter (MMD) is derived from the Dc distribution, below and above which size the rBC mass concentration is equal (Liu et al., 2019b)." sentence not clear, modify it.

12. Line 72-73 "The bulk coating thickness (Dp/Dc) was calculated as the cubic root of the total coated BC volume weighted by the total volume of rBC." Are you sure, it is weighted? I think coating thickness is ratio of cubic root of both volume (coated and core).

13. Line 74 should be "….each BC particle…."

14. Line 82-83 the use of word "excluded" here doesn't seem right. Modify it to something like "Moreover, a multi-scattering correction factor (C-value) of 3.5, 3.2 and 2.4 at the wavelengths 370 nm, 528 nm and 880 nm, respectively were utilized to correct attenuation for the multiple light scattering effect."

15. **Section 2.3** In equation 4, Is (σabs/[rBC]pri) is based to σabs-tot. If yes, pls correct it to (σabs-tot/[rBC]pri) throughout the manuscript. If not, then mention what is σabs (It can't be σabs-BC as it doesn't not include contribution of BrC)?

16. Line 102-104 not clear, modify

17. Line 136-137 The sentence not clear

"The FTIR peaks of 1630cm$^{-1}$ and 860cm$^{-1}$ are integrated the absorption areas above the baseline."

18. Line 148-149 conjunction missing.

19. Line 149-150 "The diurnal variation exhibited strong morning and afternoon rush-hour peaks." Peaks of what? Mention it in the sentence.

20. Line 156 Grammatical error "This off-road combustion sources…"

21. Line 180-181 Difficult to understand. Revise the sentence "It will introduce considerable uncertainties to use consistent MAC or AAE to derive the absorption of BC at multiple wavelengths."

22. Line 181-182 revise it to "The MAC estimated using the measured BC core size and coatings (Fig. 2c) is thus used to derive the σabs,BC (section 2.2, shown in Fig. 2d)."

23. Line 183. Grammatical error "is showed". And add a sentence mentioning variability in σabs-BC during study period (similar to variability for MACBC).

24. Line 187-192 this whole paragraph is very confusing and hard to understand. Revise it.

25. Line 202 it should be "where a1 to a5 represents the regression coefficients for each factor." ao is intercept. Modify accordingly.

26. Line 205-206 replace to "…..along with OOA2 in governing absorption of BrC."

27. Lines 206 and 207 replace the "high" to "substantial"

28. Line 207-209 Sentence not clear, revise it.

29. Line 230-231 revise to "In this context, a recent chamber study reported that the primary BrC from biomass burning plumes could be bleached to half of the initial absorptivity in 2-3 hours (Liu et al., 2021)."

30. Line 238 you can modify it to something like "This ageing or oxidation likely occurred through photooxidation during early afternoon and aqueous processes (high RH conditions prevail during nighttime) during nighttime (Fig. 4h)."

31. Line 246 "NO3 radical formed"?

32. Line 252 "This revealed that the whitening and darkening of BrC occurred simultaneously,"

33. Line 254 "location in the atmosphere." you mean geographical location or altitude, please clarify?

---

## Author Comment (AC1)

**Response to anonymous referee #1:**

*Main Comments:*

*1.Introduction Motivation is weak and objectives of study are not clear? Many studies (some of them carried out in Asia are given below) have assessed diurnal profile of BrC absorption and role of nitrogen in governing them. You can cite these paper and please explicitly state how your study is different from these.*

*R Satish, N Rastogi On the use of brown carbon spectra as a tool to understand their broader composition and characteristics: a case study from crop-residue burning samples. - ACS omega, 2019. https://doi.org/10.1021/acsomega.8b02637*

*R Satish, P Shamjad, N Thamban, S Tripathi, N Rastogi Temporal characteristics of brown carbon over the central Indo-Gangetic Plain. - Environmental science & technology, 2017. https://doi.org/10.1021/acs.est.7b00734*

**Reply:** We thank reviewer to point this out. The mentioned references are added and discussed.

"==Previous studies (Satish et al., 2017; Satish and Rastogi, 2019) found nitrogenous compounds from biomass burning were responsible for brown carbon over South Asia and the chromophores were photobleached in the afternoon.=="

L37-38

*2.Section 2.3 Citation for equation 3 and 4 missing? Many previous studies have used primary species, e.g., EC, K+ etc. for quantifying primary and secondary OC. The author can cite those papers. Moreover, relevance or applicability of assumptions taken in eq. 3 and 4 for the site are missing. Please add a brief discussion about all these aspects.*

**Reply:** The references are now added to explain the application of the minimum-R squared approach to derive the absorption of primary OA associated with BC. This method has been used in urban or sub-urban environment thus is applicable for our study.

"==Here an assumption is made that light absorption from primary aerosols is all from combustion sources, and these sources necessarily contain BC (Wang et al., 2018).== This factor is obtained using the minimum R-squared (MRS) approach (Wu and Yu, 2016), by adjusting the factor until a minimum correlation between $\sigma_{abs,secBrC}$ and [rBC] is reached because the absorption from secondary sources are least likely to covary with that from primary sources (Wang et al., 2019). ==This method has been used in urban and sub-urban environment to obtain the primary BrC associated with combustion sources.=="

L103-104, L111-112

*Further, BrC and BC emissions from different sources are very different. For e.g., vehicular emissions are highly rich in BC, but not in BrC. For biomass burning, its vice versa. How these scenarios will impact the [σabs/[rBC]pri ratio and σabs-SOA estimation. The cluster analysis (Fig S1) and AMS results indicate that scenario is likely ( Fig. 1) at the sampling site.*

*How this will impact the overall findings of this study.*

**Reply:** We thank reviewer to point this out. We agree with reviewer that different sources will have different ratios of POA/BC. However, after careful examination, there was no sporadic event such biomass burning or local pollution events during the experimental period (as indicated by the temporal evolution of attributed OA sources in Fig, 1), we therefore consider, the sources were uniform and this ratio had not significantly varied during the one-month experimental period. The ratio obtained here therefore represents the average ratio throughout the experiment. Related discussions are added.

"Different sources may exhibit different ratios of $\left(\frac{\sigma_{abs}}{[rBC]}\right)_{pri}$, however there were no sporadic pollution events during the experimental period, uniform sources are therefore considered, and this ratio tends to represent a mean for the experiment."

L112-114

*3. Line 104-105 How did the authors account for the effect of coating thickness while calculating [σabs/[rBC]pri at different wavelengths?*

**Reply:** The MA200 directly measures absorption, and the influence of BC coating thickness on the absorption of BC is considered in section 3.2 (Fig. 2).

*4. Section 3.1. (Lines 162-167): The authors reported that "Both OOA1 and OOA2 showed nighttime peak due to the dark oxidation chemistry under high relative humidity." But this may or may not be true as boundary layer height is also lower during night compared to daytime. Moreover, nitrate radicals govern the dark oxidation chemistry. Thus, nitration of organics during nighttime is a possibility, but that was not the case for OOA1 (N/C remain unchanged). Therefore, how can you attribute increase in OOA1 during night to dark oxidation chemistry? Please elaborate.*

**Reply:** We thank reviewer to point this out and have revised related discussions.

"Notably, OOA2 had a substantially higher N/C than other factors (N/C=0.037), and had highest correlation with nitrate ($r$=0.77) and with $C_xH_yN_z$ and $C_xH_yN_zO_p$ fragments ($r$=0.83). This factor therefore tends to largely result from nitrogen-containing OA and its elevation at night may be also associated with dark oxidation by nitrate radical."
"The slight enhancement at noon for OOA1 (also for OOA2) soon after morning rush-hour indicated the likely rapid formation of SOA through photooxidation. This significantly higher mean OOA2 than median value in the diurnal pattern indicated that this OA type was largely associated with pollution events. Both OOA1 and OOA2 showed nighttime peak maybe due to reduced boundary layer."
L201-202, L208-209

*5. Line 190-191. How did you come with these numbers? Please mention it probably in*

*Methodology. If these are based on σabs values, then don't use words such as "mean*

*contribution of absorption for BC, primary BrC and secondary BrC" as σabs values were not weighted with corresponding solar flux values. Instead, you can use words such as "mean contribution of absorption coefficient for BC, primary BrC and secondary BrC". Please keep this point in mind throughout the manuscript.*

**Reply:** We have rephased the absorption as absorption coefficient at appropriate places throughout the texts.

"The mean contribution of absorption coefficient for BC, primary BrC and secondary BrC is 51%, 27% and 22% in this study."

"The absorption coefficient of secondary BrC, the absorption not contributed by primary sources, is obtained by subtracting the absorption of all primary sources from the total absorption (Crilley et al., 2015)"

*6. Discussion about some figures is missing in text, e,g, Fig. 4a*

**Reply:** Related discussions are now added in section 3.4:

"The diurnal variation of $\sigma_{abs,375}$ for BC and primary BrC and their fractions showed consistent morning rush-hour peaks at 6:00-8:00 and the night-time enhancement due to reduced boundary layer (Fig. 4a-b)."

*7. Line 229-230 and elsewhere: It is mentioned that "photobleaching process causing the decreased absorption efficiency per unit mass for primary BrC." But authors have not provided any discussion about MAC or absorptivity of BrC throughout the manuscript. It is absorption coefficient they are talking about. Please careful look into it.*

**Reply:** We have added a new plot about absorption coefficient per unit mass of POA, to indicate the daytime photobleaching process.

"Fig. 4b showed the decrease of primary BrC absorption tended to be more rapid than the HOA and BBOA mass (even a slight increase for HOA, Fig. 1m and Fig. 1o), leading to decreased absorption coefficient per unit mass of primary BrC (shade in Fig. 4b), which indicates the photobleaching process."

L277-278

[Figure]

**Figure 4.** Diurnal variations of absorption coefficient at λ=375nm ($\sigma_{abs,375}$) for BC (a), primary BrC and the ==absorption efficiency of primary BrC ($\sigma_{abs,priBrC}$)/POA is shown in shade== (b), and secondary BrC, along with the $C_xH_yN_z$ and $C_xH_yN_zO_p$ fragments (c); the respective fraction in total for the segregated $\sigma_{abs,375}$ (d-f), with direct radiation shown in shade. In each plot, the lines, dots and whiskers denote the median, mean and the 25th/75th percentiles at each hour respectively.

*8. Line 250 "Overall, by apportioning the absorption of primary and secondary BrC, we found the photooxidation led to an enhanced contribution of secondary BrC by 30% but reduced contribution of primary BrC about 20% in the semi-urban environment." How did you come up with these numbers, discuss in either methodology or supplementary.*

**Reply:** The related discussions are added.

=="Fig 4e-f shows the photooxidation led to an enhanced contribution of secondary BrC by 30% but reduced contribution of primary BrC about 20%."==

L284-286

***Minor Comments:***

*1. Line 27. This sentence looks confusing. I will suggest to replace the word "shortwave absorption" to "anthropogenic absorption" or "anthropogenic radiative forcing"*

*2. Line 37: Replace "A range" to "Numerous"*

**Reply:** These are revised,

*3. Line 39-40: "which may depend on the concentration of ambient hydroxyl radical (Wang et 40 al., 2014)". This is only partially correct. Recently, some studies have reported substantial role of atmospheric condition (RH and temperature, viscosity etc.) on photochemical oxidation. For example.*

*Emerging investigator series: heterogeneous OH oxidation of primary brown carbon aerosol: effects of relative humidity and volatility, 10.1039/D0EM00311E, Environ. Sci.: Processes Impacts, 2020, 22, 2162-2171*

*Please modify the sentence and cite them properly.*

**Reply:** This is now revised.

"which may depend on the concentration of ambient hydroxyl radical (Wang et al., 2014), also influenced by relative humidity and particle volatility (Schnitzler et al., 2020)."

L41-42

*4. Line 36-40: The references cited didn't use absorptivity for half-life calculation. All these studies used BrC absorbance to indicate bleaching and BrC lifetime calculation. Please modify your sentence accordingly*

**Reply:**

"with lifetime ranging from a few hours (Zhao et al., 2015; Liu et al., 2021) to a few days (Forrister et al., 2015)"

*5. Line 43-45: revise it to something like "The enhancement of BrC absorptivity could occur either through nitration of exiting chromophores, or formation of new secondary organic aerosol (SOA) chromophores through gas-phase oxidation"*

**Reply:** This is revised.

"The enhancement of BrC absorptivity could occur either through nitration of exiting chromophores, or formation of new secondary organic aerosol (SOA) chromophores through gas-phase oxidation."

L45-47

*6. Line 48 "rule out" doesn't suit here. Replace it*

**Reply:**

"However, both processes have been rarely investigated in the field to explicitly determine the BrC components which principally determine the respective enhancement or decrease of its absorptivity, particularly in regions influenced by combined anthropogenic sources."

*7. Line 64 Grammatical error, should be "ambient aerosols were"*

**Reply:** Revised.

*8. Line 64-65 sentence not clear, revise it.*

**Reply:** We thank reviewer to point this out and we have revised:

"In this study, the ==ambient aerosols were== sampled ==by a large-flow (1.05 m$^3$ min$^{-1}$) air particle sampler (TH-1000C II) with a PM$_{2.5}$ impactor (BGI SCC 1.829)== and dried by a silica drier before measurement."

L67-68

*9. Line 65 should be "…..refractory black carbon (rBC) mass."*

**Reply:** Revised.

*10. Line 66-68 Add a little bit more detail in this context.*

**Reply:**

The single particle soot photometer (SP2, DMT., USA) ==used continuous laser at λ=1064nm to incandesce light-absorbing aerosols (such as BC) for irradiating detectable visible light.== The incandescence signal was used to measure the ==refractory black carbon (rBC) mass==."

L69-70

*11. Line 71-72 "The mass median diameter (MMD) is derived from the Dc distribution, below and above which size the rBC mass concentration is equal (Liu et al., 2019b)." sentence not clear, modify it.*

**Reply:** This is revised.

"The mass median diameter (MMD) is derived from the Dc distribution, ==which is determined as== below and above ==MMD== the rBC mass concentration is equal (Liu et al., 2019b)."

*12. Line 72-73 "The bulk coating thickness (Dp/Dc) was calculated as the cubic root of the total coated BC volume weighted by the total volume of rBC." Are you sure, it is weighted? I think coating thickness is ratio of cubic root of both volume (coated and core).*

**Reply:** This is revised.

"The bulk coating thickness (*Dp/Dc*) is calculated as ==the cubic root of ratio of the total coated BC volume divided by the total volume of rBC.=="

L78

*13. Line 74 should be "....each BC particle...."*

**Reply:** Revised.

*14. Line 82-83 the use of word "excluded" here doesn't seem right. Modify it to something like "Moreover, a multi-scattering correction factor (C-value) of 3.5, 3.2 and 2.4 at the wavelengths 370 nm, 528 nm and 880 nm, respectively were utilized to correct attenuation for the multiple light scattering effect."*

**Reply:** We thank reviewer to point this out and we have revised:

"==Moreover, a multi-scattering correction factor (C-value) of 3.5, 3.2 and 2.4 at the wavelengths 370 nm, 528 nm and 880 nm, respectively were utilized to correct attenuation for the multiple light scattering effect.=="

L89-91

*15. Section 2.3 In equation 4, Is ($\sigma$abs/[rBC]pri) is based to $\sigma$abs-tot. If yes, pls correct it to ($\sigma$abs-tot/[rBC]pri) throughout the manuscript. If not, then mention what is $\sigma$abs (It can't be $\sigma$abs-BC as it doesn't not include contribution of BrC)?*

**Reply:** We thank reviewer to point this out. $\sigma_{abs}$ is now revised $\sigma_{abs-total}$.

*16. Line 102-104 not clear, modify*

*17. Line 136-137 The sentence not clear "The FTIR peaks of 1630cm-1 and 860cm-1 are integrated the absorption areas above the baseline."*

*18. Line 148-149 conjunction missing.*

**Reply:** These are revised.

*19. Line 149-150 "The diurnal variation exhibited strong morning and afternoon rush-hour peaks." Peaks of what? Mention it in the sentence.*

**Reply:** Revised.

"The diurnal variation exhibited strong morning and afternoon rush-hour peaks ==of mass concentration.=="

*20. Line 156 Grammatical error "This off-road combustion sources..."*

*21. Line 180-181 Difficult to understand. Revise the sentence "It will introduce considerable uncertainties to use consistent MAC or AAE to derive the absorption of BC at multiple wavelengths."*

*22. Line 181-182 revise it to "The MAC estimated using the measured BC core size and*

*coatings (Fig. 2c) is thus used to derive the σabs,BC (section 2.2, shown in Fig. 2d)."*

**Reply:** These are revised.

*23. Line 183. Grammatical error "is showed". And add a sentence mentioning variability in $\sigma$ abs-BC during study period (similar to variability for MACBC).*

**Reply:** Revised.

The $\sigma_{abs,BC}$ was 9.1±7.3 Mm$^{-1}$ during experimental period. MAC of BC at $\lambda$=375nm showed to be at 8.4 -16.6 m$^2$ g$^{-1}$ with enhanced absorption when high coatings."

L225

*24. Line 187-192 this whole paragraph is very confusing and hard to understand. Revise it.*

**Reply:** We thank reviewer to point this out and we have revised:

"Using the method above, the total ($\sigma_{abs,total}$) and attributed absorption of BC ($\sigma_{abs,BC}$), primary ($\sigma_{abs,priBrC}$) and secondary BrC ($\sigma_{abs,secBrC}$) at $\lambda$=375nm are shown in Fig. 3a-c. In Fig. 3b, the brown and green shades above the adjacent tracer indicate the absorption coefficient of primary and secondary BrC, respectively. Fig. 3c shows that the absorption coefficient of primary BrC was higher than secondary BrC for most time, but for certain periods they were equivalent or secondary BrC occasionally exceeds primary BrC. The mean contribution of absorption coefficient for BC, primary BrC and secondary BrC is 51%, 27% and 22% in this study. The tracers associated with nitrogen-containing organics, such as OOA2 (with highest N/C), $C_xH_yN_z$ and $C_xH_yN_zO_p$ fragments, and the FTIR measured -NO + -NO$_2$, are also shown in Fig. 3d-e."

L230-234

*25. Line 202 it should be "where a1 to a5 represents the regression coefficients for each factor." ao is intercept. Modify accordingly.*

*26. Line 205-206 replace to ".....along with OOA2 in governing absorption of BrC."*

*27. Lines 206 and 207 replace the "high" to "substantial"*

*28. Line 207-209 Sentence not clear, revise it.*

*29. Line 230-231 revise to "In this context, a recent chamber study reported that the primary BrC from biomass burning plumes could be bleached to half of the initial absorptivity in 2-3 hours (Liu et al., 2021)."*

*30. Line 238 you can modify it to something like "This ageing or oxidation likely occurred through photooxidation during early afternoon and aqueous processes (high RH conditions prevail during nighttime) during nighttime (Fig. 4h)."*

**Reply:** These are revised.

*31. Line 246 "NO3 radical formed"?*

**Reply:** This is revised.

"The nighttime chemistry involving $NO_3$ radical through the oxidation of $NO_2$ by $O_3$,"

*32. Line 252 "This revealed that the whitening and darkening of BrC occurred simultaneously,"*

*33. Line 254 "location in the atmosphere." you mean geographical location or altitude, please clarify?*

**Reply:** These are revised.

---

## Author Comment (AC2)

**Response to anonymous referee #2:**

*Main Comments:*

*1.The identification of OA sources with PMF analysis could be improved. The authors could analyze how the factor mass spectra identified in the present study correlate with previous results. A library with existing profiles can be found here:* [https://cires1.colorado.edu/jimenez-group/HRAMSsd/](https://cires1.colorado.edu/jimenez-group/HRAMSsd/).

*In addition the mass spectra can be compared to unit mass resolution reference spectra from* [https://cires1.colorado.edu/jimenez-group/AMSsd/](https://cires1.colorado.edu/jimenez-group/AMSsd/).

**Reply:** We thank reviewer to point this out. All resolved factors are now compared with literatures and the library reviewer suggested. Related discussions are now added.

[revised manuscript text omitted]

L166-168, L169-175, L176-178, L182-188, L195-199, L203-204

*In addition, COA in previous works usually shows a peak at noon, while in this study the lunch peak is barely visible. The author should discuss this discrepancy.*

**Reply:** The was only a minor peak at noon for COA, which may be due to the sub-urban nature of the site where the major aerosols from cooking sources may have been processed and lost the signature near source. Related discussions are now added:

"The was only a minor peak at noon for COA, which may be due to the sub-urban nature of the site where the major aerosols from cooking sources may have been processed and lost the signature near source. The feature of this factor was also observed in sub-urban environment (Huang et al., 2021)."

L179-181

*Finally, the authors claim the use of external tracers to identify the PMF factors, but for COA an internal tracer was used instead, which makes the attribution risky, especially considering the correlation in time with HOA factors (based on the diurnal profile).*

**Reply:** Previous literatures have widely used $C_6H_{10}O^+$ is considered a signature fragment mainly from cooking emission rather than from traffic (Sun et al., 2011b), but an unambiguous external tracer for cooking source is difficult to find. We have tested $C_6H_{10}O^+$ had a much weaker correlation with HOA (r = 0.48) than COA (r = 0.80), thus this factor is likely COA rather than HOA. In addition, the correlation between HOA and COA is 0.31 for time series, and 0.42 for mass spectra, therefore these factors can be discriminated.

*2. This study identifies organic nitrate using ATR-FT-IR, integrating the spectra area around the characteristic absorption peaks at 860 cm⁻¹ and 1640 cm⁻¹, in agreement with Liu et al. (2012). Nevertheless previous studies showed that the region between 1600 and 1700 cm⁻¹*

*shows typically a strong absorption signal due to the carbonyl group of ketones and carboxylic acid (Maria et al., 2002; Russell et al., 2009), which would lead to an overestimation of the NO₂ absorption at 1640 cm⁻¹.*

**Reply:** We thank reviewer to point this out. Although the carbonyl group has absorption at 1640 cm$^{-1}$- 1850 cm$^{-1}$ (Russell et al., 2009), and Maria et al. (2003) pointed out the absorption peak of carbonyl group was around 1720 cm$^{-1}$. However there was no discernable peak of carbonyl group for our infrared spectrum, and the peak of OH at 2500 cm$^{-1}$ - 3400 cm$^{-1}$ for the carboxylic acid is not discernable neither, thus the influence of ketone and carboxylic acid may be of less importance for our dataset. The related discussions are added.

"There was no discernable peak of carbonyl group for our infrared spectrum, and the peak of OH at 2500 cm$^{-1}$ - 3400 cm$^{-1}$ for the carboxylic acid is not discernable neither, thus the influence of ketone and carboxylic acid may be of less importance for our dataset."

L153-155

*3. The discussion about the bleaching and darkening of BrC is based on the analysis of*

*diurnal profiles of primary and secondary BrC , both absolute absorption coefficient and fractional contribution in figure 4. The text reports: "Fig. 4b showed the decrease of primary BrC absorption tended to be more rapid than the HOA and BBOA mass (even a slight increase for HOA), which indicated the likely photobleaching process", but this decrease is difficult to discern in the figure.*

**Reply:** A new plot about absorbing efficiency (absorption coefficient divided by mass) is now added in Fig. 4b to aid this conclusion. The related discussions are revised.

"Fig. 4b showed the decrease of primary BrC absorption tended to be more rapid than the HOA and BBOA mass (even a slight increase for HOA, Fig. 1m and Fig. 1o), leading to decreased absorption coefficient per unit mass of primary BrC (the shade in Fig. 4b), which indicates the photobleaching process."

L277-278

[Figure]

**Figure 4.** Diurnal variations of absorption coefficient at λ=375nm ($\sigma_{abs,375}$) for BC (a), primary BrC and the absorption efficiency of primary BrC ($\sigma_{abs,priBrC}$)/POA is shown in shade (b), and secondary BrC, along with the $C_xH_yN_z$ and $C_xH_yN_zO_p$ fragments (c); the respective fraction in total for the segregated $\sigma_{abs,375}$ (d-f), with direct radiation shown in shade. In each plot, the lines, dots and whiskers denote the median, mean and the 25[th]/75[th] percentiles at each hour respectively.

*In addition, the attribution of secondary BrC to local photochemical production is based on the comparison between the fraction of secondary BrC diurnal profile and solar radiation, but if local photochemistry triggered secondary BrC formation I would expect to see a correlation between secondary BrC absorption (reported in fig.4 c) and solar radiation. On the contrary, secondary BrC absorption shows a peak in the morning, when photochemistry is expected to be lower.*

**Reply:** We have carefully considered the comments from reviewer. The morning peak coinciding with the primary BrC can be explained as the rapid formation of BrC from sources when emitted gases rapidly condensed and formed aerosols. These may lead to high cooccurrence between primary and secondary BrC. Previous studies in urban environment also observed concurrent peaks of primary and secondary BrC, which usually occurred at morning rush hour (Zhang et al., 2020). In addition to the morning rush-hour peak, a peak after midday also observed for secondary BrC, and this small peak at noon was consistent with the peak of solar radiation, confirming that local photochemistry triggered the formation of secondary brown carbon. Related discussions are revised.

"The morning peak coinciding with the primary BrC may result from the rapid formation of BrC from sources when emitted gases condensed and formed aerosols. These may lead to high cooccurrence between primary and secondary BrC. Previous studies in urban environment also

observed concurrent peaks of primary and secondary BrC, which usually occurred at morning rush hour (Zhang et al., 2020)."

L271-274

***Minor Comments:***

*1.Line 34-36. Please revise this sentence. Saleh et al. 2014 reported that the OA to BC ratio is higher during the smoldering phase, but do not compare the absorption efficiency of BrC produced during smoldering and flaming. Similarly, Chakrabarty et al. observed an increase in the absorption angstrom exponent of aerosol particles during smoldering, due to the larger OA contribution, but did not report differences in the imaginary part of the BrC refractive index during smoldering and flaming.*

**Reply:** We have revised this sentence according to reviewer's suggestion:

"These primary BrC had a range of absorptivity, which was found to be controlled by burning phases, with OA co-emitting with BC (the flaming phase) exhibiting a higher absorptivity than OA-dominated smoldering phase (Liu et al., 2021)."

L34-36

*2.The authors classify the sampling period based on the analysis of back-trajectories (see figure S1). The sampling site is located in a suburban area of Beijing where local and nearby pollution sources are likely affecting the observed PM trend, rather than synoptic scale circulation. If the author wants to discriminate the sampling period into cluster, I would suggest to use local meteorology, including temperature, relative humidity, and wind speed/direction. For example, figure S1 shows an increase in the concentration and relative contribution of nitrate when relative humidity is higher, suggesting the relevance of local processes. Furthermore, wind speed and direction might help to spot the time when the impact of the urban Beijing area is higher.*

**Reply:**

We thank reviewer to point this out. Local meteorology including wind and RH is also examined, which was found not to be the main driving factor to determine the pollution level, but the synoptic circulation of air mass is the major factor. This is because Beijing city acts strict environmental regulations but the air pollutants were synoptically transported from the polluted southern regions to the Beijing City, while the rapidly transported cleaner air from the north usually diluted the pollutants. The results here are consistent with a wide range of previous studies about the pollution conditions associated with synoptic patterns in Beijing region (Wu et al., 2022; Liu et al., 2019; Hu et al., 2020).

We have also included the local wind information in the revised figure S1.

[Figure]

**Figure S1. (a) Clustered back-trajectories for the past 72 hours during the experiment with markers denoting 12h intervals. (b-d) Statistics for the concentrations of key aerosol compositions from each cluster. The whiskers, box boundaries and lines in box denote the 10th/90th percentiles, 25th/75th percentiles and the median, respectively. (e) Time series of RH and T, (f) wind speed colored by wind direction, (g) mass concentrations of key aerosol compositions.**

*3.Line 77: the authors derived BC MAC based on the Mie theory. Liu et al 2018 showed that the Mie theory holds for spherical particles, but fails in reproducing the absorption of fractal particles. The author should discuss the uncertainty derived from it.*

**Reply:** The coatings obtained by the SP2 measurement at λ=1064nm can be relatively independent of particle shape owing to the longer measurement wavelength, as discussed in previous studies (Liu et al., 2014; Hu et al., 2021). Related discussions are added.

"The SP2 measurement at λ=1064nm longer than mostly populated BC size means the derived coatings and subsequent calculation of MAC is relatively independent of particle shape within uncertainty of 21% (Liu et al., 2014; Hu et al., 2021)."

L83-85

*4.Line 105: How do the primary absorption to rBC concentration ratios compared with previous studies?*

**Reply:**

"The $\left(\frac{\sigma_{abs,total}}{[rBC]}\right)_{pri}$ ratio at λ=375 nm, 470 nm, 528 nm, 635 nm and 880 nm is calculated to

be 20.7, 17.0, 14.4, 11.7 and 5, respectively (Fig. S2), which falls within the reported values from previous studies 11-50 (Wang et al., 2019; Zhang et al., 2020)."

L117-118

*5.Line 141-142: Inorganic nitrate usually dominates nitrate signal in the AMS measurements (Farmer et al., 2010). Please, revise this sentence or estimate organic nitrate from AMS signal and compare it with inorganic nitrate.*

**Reply:** This sentence has been removed.

***Technical comments:***

*Line 83: corrected instead of excluded*

*Line 181: constant instead of consistent*

*Line 226: accounted for instead of occupied*

**Reply:** These are now corrected.

*Figure1: the author might want to change the order of factors in the figure and report*

*OOA1 before OOA2.*

**Reply:** This is revised.

*Figure 3: If possible, the author might want to use horizontal lines instead of circles as markers in fig 3e to clarify that the FTIR data correspond to a time range of 24 hours. The horizontal lines should start and end at the beginning and end of the corresponding sampling period.*

*Figure 3: panel b shows a day dominated by secondary BrC at the end of the field experiment (likely May 7), while in fig 3c the secondary BrC absorption during the same day is not reported.*

**Reply:** We thank reviewer to point this out. In Fig. 3, the missing data is now corrected and markers are revised according to reviewer's suggestion.

[Figure]

**Figure 3. Temporal evolution of segregated absorbing properties. (a) Absorbing coefficients ($\sigma_{abs}$) at multiple wavelengths measured by the aethalometer, (b) $\sigma_{abs}$ at $\lambda$=375nm ($\sigma_{abs,375}$) for all aerosols, primary OA and BC, (c) $\sigma_{abs,375}$ for primary BrC and secondary BrC. (d) mass concentration of OOA2 and the $C_xH_yN_z$ and $C_xH_yN_zO_p$ fragments measured by the AMS. (e) FTIR-measured absorption of -NO and -NO₂ bonds.**

---

## Referee Report (RR1)

Referee report on ACP-2022-483 - Concurrent photochemical whitening and darkening of ambient brown carbon

Qian Li[1], Dantong Liu[1*], Xiaotong Jiang[1], Ping Tian[2], Yangzhou Wu[1], Siyuan Li[1], Kang Hu[1], Quan Liu[3], Mengyu Huang[2], Ruijie Li[2], Kai Bi[2], Shaofei Kong[4], and Deping Ding[2]

The journal article on "Concurrent photochemical whitening and darkening of brown carbon" by Li et al., describes the behavior of primary and secondary brown carbon (BrC) in a sub-urban site near Beijing, China. By apportioning the total aerosol absorption between black carbon (BC), primary BrC and secondary BrC, they identify that traffic and biomass burning are main sources of primary BrC and that nitrogen-containing moderately oxygenated organic aerosol are the main source of secondary BrC. Further, a percentage decrease in primary BrC is observed together with a percentage increase in absorbance by secondary BrC from photooxidation, which is considered to offer field evidence of the concurrent of whitening and darkening of BrC.

Overall, there are some interesting methodologies used to separate primary and secondary BrC and use their diurnal variation to illustrate the dynamic behavior of BrC in the atmosphere. However, there is a significant lack of discussion on the various possible interpretations of the results, other than the authors' main conclusions. There is also a serious lack of discussion on uncertainties associated with the measurements/calculations. Furthermore, a language revision may be required to express the main findings in a more clear and concise manner. However, as extensive measurements from various instruments are available to them, a revised version of this manuscript that addresses these issues may be considered after review.

**Main Comments**

1. The method used to apportion BrC absorbance to primary and secondary BrC, described by Wang et al. (2019), uses the assumption of a constant $(\sigma_{abs,total}/[rBC])_{pri}$. Previous reviewers have also raised their concerns regarding the validity of this assumption. While the authors have said that there are no pollution events that may result in a change in these values, the Figures 1 and 2 show that there may have been a few instances of such events. The authors also discuss this possibility in Lines 219-220 when discussing the changes observed for BC coating thickness during high pollution events. While it may not be straightforward to use a $(\sigma_{abs,total}/[rBC])_{pri}$ that is composition dependent, it is worth to mention the uncertainty associated with the assumption. For example, is it possible to discuss how the range of values for this ratio can affect the final BrC calculations?

   I believe a thorough investigation on the uncertainties associated with this ratio and the other components used equations are necessary. Also, a propagated error calculation can add value to the final output of these calculations.

2. As the Micro Aethalometer gives absorbance measurements in three other wavelengths where BrC absorbance may be observed (i.e. 470,528 and 635nm), have the authors observed the same behavior of BrC in these wavelengths as well? Is the relative contribution or diurnal profile of primary and secondary BrC any different? Furthermore, including calculations of wavelength dependence, which is another important parameter that can describe the absorbing properties, can enrich the discussion.

3. Line 248- is r>0.4 considered as a high correlation?

4. Line 226 -  Do the MAC values of BC match well with those in literature?

5. Lines 277-278 – The authors use the comparison of diurnal profiles, the reduction in the absorbance and the absorption coefficient per unit POA mass, as strong evidence of photobleaching of primary BrC. While photobleaching maybe one of the reasons for the observed reduction, the given evidence is not strong enough to make this an absolute conclusion. It is important to discuss if there are other possibilities for this. For an example, it possible that some primary species transformed into less absorbing secondary BrC species, and this could be via other reaction pathways. The type of HOA/BBOA contributing to the absorption during this period may have lower absorptivity. While it may be difficult to provide solid evidence for all the claims, it is important to discuss the possible reasons for such observations.

   Also, considering that BBOA is more absorbing per unit mass than traffic-related OA and has a corresponding peak at night, it is surprising to see the low, almost constant levels of abs/POA from 6-9pm. I believe the reasons for this have not been discussed in the manuscript.

   Furthermore, while the method used to apportion primary and secondary BrC has been previously used, it may be important to point out that it may not be as straightforward to separate primary and secondary sources of BrC by assuming that all primary BrC are from combustion sources and that there is no cooccurrence of primary and secondary BrC from these. In fact, authors themselves mention that their study and those previously do observe a cooccurrence of these emissions (Lines 272-273). Therefore, it is possible that some of the primary sources are being attributed to secondary sources and vice versa. This maybe a possible reason for the simultaneous peak observed for primary and secondary BrC during morning rush hour.

6. I am unable to follow how the authors determined that there is a 20% decrease in primary and 30% increase in secondary BrC absorbance due to photooxidation. If I understand correctly, it says in the methodology that this is compared to the overall average absorbance of primary and secondary BrC absorbance. If this is the case, how is it determined that photooxidation alone was responsible for the increase or decrease when there are various processes taking place throughout the day that affect BrC absorbance. I notice that previous reviewers have also raised this concern, but I don't believe it has been properly addressed in the revised manuscript.

7. Figure 4 (d-e) –Authors may consider to use "Fraction of total absorbance" or a similar title that better describes what the axis represents
   This is the case throughout the text where, when a fraction or a percentage is used. Ideally, the description should include the denominator. It may be particularly important to describe it properly when the percentage decrease and increase of primary and secondary BrC absorbance is used to describe the effect of photochemical reactions (Lines 22-23 and 300-301) as this is one of the primary claims.

**Minor comments**

(i)     The sentences in the abstract are too long and the message is confusing. Please rephrase. For example;

" The absorption of BC is constrained by its size distribution and mixing state, being subtracted from total absorption to obtain the absorption of BrC, then by applying the least-correlation of BC

absorption with secondary BrC, the absorption contributed by BC, primary BrC and secondary BrC was apportioned" can be rephrased as;

"The absorption of BC is constrained by its size distribution and mixing state and the BrC absorption is obtained by subtracting the BC absorption from the total aerosol absorption. Aerosol absorption was further apportioned to BC, primary BrC and secondary BrC by applying the least-correlation between secondary BrC and BC."

Several other instances where the messaging is unclear due to long sentences can be found throughout the manuscript. Please try to write in clear and concise sentences to get the message across more clearly.

(ii)     Several grammar, spelling mistakes and missing words can be found. Please read thoroughly to minimize the errors. I am only listing a few examples.

e.g.: Line 27 - Atmospheric absorbing organic aerosol (OA), known as brown carbon (BrC), is "a" important contributor to anthropogenic 28 absorption besides black carbon (BC)

Line 34 – These primary BrC "has" ….(again long sentence here and the message is unclear).

Line 46- "Existing" chromophores

(iii)    The word brown carbon used in the middle of text after acronym BrC is first introduced. Please be consistent.

(iv)    Line 37 - Dasari, Sanjeev, et al. "Photochemical degradation affects the light absorption of water-soluble brown carbon in the South Asian outflow." Science advances 5.1 (2019) also discuss photochemical degradation of South Asian outflow.

(v)     Line 39  - "decease" should be changed to "decrease" ….and photobleaching "of"

---

## Referee Report (RR2)

Referee report on ACP-2022-483 - Concurrent photochemical whitening and darkening of ambient brown carbon

Qian Li[1], Dantong Liu[1*], Xiaotong Jiang[1], Ping Tian[2], Yangzhou Wu[1], Siyuan Li[1], Kang Hu[1], Quan Liu[3], Mengyu Huang[2], Ruijie Li[2], Kai Bi[2], Shaofei Kong[4], and Deping Ding[2]

The manuscript by Li et al. titled "Concurrent photochemical whitening and darkening of brown carbon" describes the behavior of primary and secondary brown carbon (BrC) from field observation in a sub-urban site near Beijing, China. The total aerosol absorption is apportioned between black carbon (BC), primary BrC and secondary BrC. Traffic and biomass burning are identified as main sources of primary BrC and nitrogen-containing oxygenated organic aerosol are identified as the main source of secondary BrC. A reduction is observed in primary BrC absorbance/total absorbance during the day time with a simultaneous increase in secondary BrC absorbance/total absorbance. Their main finding that there is field evidence of concurrent whitening and darkening of BrC due to photochemical processes, is primarily based on this observed diurnal variation.

While there is some improvement from the previous version with the inclusion of other possible explanations for the observations and in acknowledging the uncertainties in the used method, it is questionable whether the authors have provided sufficient evidence that justify their main findings are indeed evidence of whitening and darkening of BrC, rather than formation and degradation of BrC.

**Main Comments**

1. In Figure 4b), authors have shown the reduction in absorption coefficient of primary BrC/POA (i.e. the MAC of POA) as evidence for the whitening of primary BrC. While there are many uncertainties and other possibilities for this, it is offering some level of evidence for the "whitening" (a better term may be degradation) of primary BrC. However, the same is not shown for secondary BrC. The term "darkening" indicates that the formed secondary BrC increased their absorbance. The increase in secondary BrC abs/total abs only indicates that the absorbance from secondary BrC increased during this time. Due to the very obvious formation of SOA from photochemical processes will indeed result in this. However, it does not indicate that there was a "darkening" process. What does the diurnal variation of MAC of SOA look like? Are you able to see an increase in MACSOA? As OA factors from PMS and BrC absorbance (and primary and secondary BrC absorbance) is available, is it possible to investigate the MAC of the individual OA factor? Do their diurnal variations offer additional information of the "whitening and darkening" effect.

2. Lines 269-271-While the overall AAE of total absorbance may be highly influenced by BC absorbance, the AAE of total BrC, primary BrC and secondary BrC can be individually calculated as the total absorbance has already been apportioned.

3. Line 291-293 – The meaning is very unclear

4. Line 22 and line 316- "fraction of total absorbance of secondary BrC" is unclear. Do you mean fraction of absorbance of secondary BrC to total absorbance?

5. Lines 327-336 - As the primary focus of the paper is the "whitening and darkening" of BrC, a more detailed analysis with further evidence is required to make a strong conclusion on this. The 4. Conclusions section does not even include this main point and does not offer any atmospheric implication of the major observations from this manuscript.

---

## Referee Report (RR3)

Referee report on ACP-2022-483 - Concurrent photochemical whitening and darkening of ambient brown carbon

Qian Li[1], Dantong Liu[1*], Xiaotong Jiang[1], Ping Tian[2], Yangzhou Wu[1], Siyuan Li[1], Kang Hu[1], Quan Liu[3], Mengyu Huang[2], Ruijie Li[2], Kai Bi[2], Shaofei Kong[4], and Deping Ding[2]

The manuscript titled "Concurrent photochemical whitening and darkening of brown carbon" by Li et al. describes the behavior of primary and secondary brown carbon (BrC) from field observation in a sub-urban site near Beijing, China. The main finding of this work is that it is offering field evidence of concurrent whitening and darkening of ambient BrC.

The reviewers and editor have raised several concerns and have given suggestions for improvement, most of which have been addressed by the authors. However, it appears that the authors' have not fully grasped the reasoning for these comments and have failed to modify the manuscript in a significant way. Some of the comments have been addressed by adding one or two sentences in the suggested paragraph and the changes are not reflected anywhere else in the manuscript, even if it changes some of the main messaging. Take for example the comment by editor that additional evidence to justify the main claim of concurrent whitening and darkening, other than the percentage difference in primary and secondary BrC absorption, is needed. This has only been addressed in one line in the accompanying text, suggesting the enhancement in MAC of SOA, but not addressed in the abstract, where it only describes the percentage difference.

Furthermore, comments have been made regarding calculating the MAC of individual OA factors, and has not been addressed. With source-apportioned OA factors and total OA absorbance, several studies have used methods such as multiple linear regression analysis (Qin et al. "Chemical characteristics of brown carbon in atmospheric particles at a suburban site near Guangzhou, China." *ACP* 18.22 (2018): 16409-16418., Wang et al. "Wintertime optical properties of primary and secondary brown carbon at a regional site in the North China Plain." *ES&T* 53.21 (2019): 12389-12397. Kasthuriarachchi et al. "Light absorbing properties of primary and secondary brown carbon in a tropical urban environment." *ES&T* 54.17 (2020): 10808-10819., Wang et al. "Aqueous production of secondary organic aerosol from fossil-fuel emissions in winter Beijing haze." *PNAS* 118.8 (2021): e2022179118. and latest studies with ridge regression models (Zhang et al. "Impact of COVID-19 lockdown on the optical properties and radiative effects of urban brown carbon aerosol." *Geo. Front.* 13.6 (2022): 101320.) to obtain MAC of individual OA factors. The fact that authors have failed to do the same is questionable. The addition of this information has the potential to offer supporting evidence to the main claim of this work as it will help to identify how the MAC of each OA factor changed during the day.

A complete revision of this work, including MAC of individual factors and addressing the uncertainties of the calculation, assumptions etc., with more focus on the light absorbing properties in the Results and Discussion section (more than half is attributed to OA source discussion), with a thorough language revision may be re-submitted for revision.

---

## Referee Report (RR4)

Referee report on ACP-2022-483 - Concurrent photochemical whitening and darkening of ambient brown carbon

Appreciate the effort of the authors to address the comments in detail and incorporate them in the necessary content throughout the manuscript. I believe the manuscript is suitable for publication after the following minor comments are addressed. Overall, as there are several instances of long sentences and grammatical errors that cloud the important messaging, a thorough language revision may be considered.

Minor comments;

Lines 22-25-MAC and contribution to total absorbance of POA and SOA could be discussed for POA and SOA separately to make the message clearer. For example. "Photochemical processes were found to reduce the mass absorption cross section (MAC) of primary OA, reducing its contribution to total absorption by 20%, at the same time increasing MAC for secondary OA, which showed a 30% enhancement in contribution to total absorbance….."

Line 25-26-"The study provides……..*nitrogen-containing secondary* OA can compensate for some effects of bleaching of primary BrC."

In some text, the authors have implied that photooxidation was responsible for the observed changes while in other instances photochemical processes are mentioned. As the study cannot isolate what type of reactions may have caused the increase or decrease in absorptivity, maybe it is safer to use the term photochemical processes. Please read through and keep the message consistent.

Line 318-321 -The sentences are not well combined. Re-writing these sentences combining the messages maybe clearer.

Line 321- Needs revision for grammatical errors and plain language.

Line 324- Can be considered to be revised as "Though other processes such as aqueous-phase reactions may cause changes to MAC of BrC at nighttime, the apparent change in aerosol absorption observed in this study during daytime can play an important role on the radiative impacts due to intensive solar radiation during daytime"?

Lines 314-339 – If possible, it may be better to revise this paragraph and re-organise the important information mentioned. The current layout appears to be jumping back and forth between MAC, relative contribution of POA and SOA and SOA formation pathways.

Line 340-342 – Sentence too long and unclear. Please re-write as two or more sentences.

Line 340-348 – Can be considered to be included in Conclusions section instead, as it is repeating some of the main messages that was described in the paragraph right above.

Lines 360-361- "These OA could primarily emit as aerosol phase, or in gas phase which requires further oxidation to be in aerosol phase to serve as BrC." -  I am unclear of the relevance of this sentence.

---

## Author Response (AR2)

Dear Editor and Reviewers,

We are thankful for the insightful comments on our manuscript. We have now addressed all comments and revised our previous manuscript accordingly. The corresponding changes in the texts are highlighted in yellow.

**Reviewer 3:**

*The journal article on "Concurrent photochemical whitening and darkening of brown carbon" by Li et al., describes the behavior of primary and secondary brown carbon (BrC) in a sub-urban site near Beijing, China. By apportioning the total aerosol absorption between black carbon (BC), primary BrC and secondary BrC, they identify that traffic and biomass burning are main sources of primary BrC and that nitrogen-containing moderately oxygenated organic aerosol are the main source of secondary BrC. Further, a percentage decrease in primary BrC is observed together with a percentage increase in absorbance by secondary BrC from photooxidation, which is considered to offer field evidence of the concurrent of whitening and darkening of BrC.*

*Overall, there are some interesting methodologies used to separate primary and secondary BrC and use their diurnal variation to illustrate the dynamic behavior of BrC in the atmosphere. However, there is a significant lack of discussion on the various possible interpretations of the results, other than the authors' main conclusions. There is also a serious lack of discussion on uncertainties associated with the measurements/calculations. Furthermore, a language revision may be required to express the main findings in a more clear and concise manner. However, as extensive measurements from various instruments are available to them, a revised version of this manuscript that addresses these issues may be considered after review.*

*Main Comments:*

*1. The method used to apportion BrC absorbance to primary and secondary BrC, described by Wang et al. (2019), uses the assumption of a constant $(\sigma_{abs,total}/[rBC])_{pri}$ . Previous reviewers have also raised their concerns regarding the validity of this assumption. While the authors have said that there are no pollution events that may result in a change in these values, the Figures 1 and 2 show that there may have been a few instances of such events. The authors also discuss this possibility in Lines 219-220 when discussing the changes observed for BC coating thickness during high pollution events. While it may not be straightforward to use a $(\sigma_{abs,total}/[rBC])_{pri}$ that is composition dependent, it is worth to mention the uncertainty associated with the assumption. For example, is it possible to discuss how the range of values for this ratio can affect the final BrC calculations?*

*I believe a thorough investigation on the uncertainties associated with this ratio and the other components used equations are necessary. Also, a propagated error calculation can add value to the final output of these calculations.*

**Reply:** We thank reviewer to point this out. We have calculated $\left(\frac{\sigma_{abs,total}}{[rBC]}\right)_{pri}$ for the high pollution period, which is 19.1% difference with the experiment mean. The uncertainty of this

parameter is mainly associated with the datapoints used according to Wang et al. (2019). According to reviewer's suggestion, we have gone through the uncertainties of each input parameter and output parameter using error propagation, which is now added in Table S1. The following is now added to discuss the uncertainties of each parameter.

"The uncertainty of $\left(\frac{\sigma_{abs,total}}{[rBC]}\right)_{pri}$ is 4% for the data points over 1.5 according to (Wang et al., 2019). The measurement of rBC mass from the SP2 had uncertainty of 20% (Schwarz et al., 2008), with relative coating thickness having uncertainty of 23% (Taylor et al., 2015), hereby resulting in a uncertainty of 27% for calculated $MAC_{BC}$. The above results in uncertainties of 31% and 20% for $\sigma_{abs,BC}$ and $\sigma_{abs,pri}$, respectively. The absorption measurement by MA200 had uncertainty of 25% ((Drinovec et al., 2015b; Duesing et al., 2019). All these uncertainties propagates the uncertainties of $\sigma_{abs,BrC}$, $\sigma_{abs,priBrC}$ and $\sigma_{abs,secBrC}$ as 40%, 37% and 32% respectively. These are summarized in Table S1."

L223-229

**Table S1. Estimated uncertainties of input and output parameters.**

| Input Parameter | Uncertainty (%) | Output Parameter | Uncertainty (%) |
|---|---|---|---|
| $\left(\frac{\sigma_{abs,total}}{[rBC]}\right)_{pri}$ | 4[a] | $\sigma_{abs,BrC}$ | 40 |
| BC mass concentration | 20[b] | $\sigma_{abs,priBrC}$ | 37 |
| MAC | 27[c] | $\sigma_{abs,secBrC}$ | 32 |
| $\sigma_{abs,BC}$ | 31 | | |
| $\sigma_{abs,pri}$ | 20 | | |
| $\sigma_{abs,total}$ | 25[d-e] | | |

(a)  Wang et al. (2019)

(b)  Schwarz et al. (2008)

(c)  Taylor et al. (2015)

(d)  Duesing et al. (2019)

(e)  (Drinovec et al., 2015)

*2. As the Micro Aethalometer gives absorbance measurements in three other wavelengths where BrC absorbance may be observed (i.e. 470,528 and 635nm), have the authors observed the same behavior of BrC in these wavelengths as well? Is the relative contribution or diurnal profile of primary and secondary BrC any different? Furthermore, including calculations of wavelength dependence, which is another important parameter that can describe the absorbing properties, can enrich the discussion.*

**Reply:** The parameters at other longer wavelengths (470, 528 and 635nm) also showed similar results with that at 375nm but with decreased fraction of BrC absorption with increased wavelength. We have also calculated and discussed AAE, which is the parameter to reflect the

wavelength dependence of absorption. Due to the relatively high contribution of BC to total absorption, we have not found apparent variation of AAE from the bulk measurement.

The related discussions are now added.

"The relative contribution and diurnal variation of primary and secondary BrC measured by MA200 at 470, 528 and 635nm wavelengths are similar to those at 375nm wavelengths, but with decreased fraction of BrC absorption with increased wavelength. Due to the high contribution of BC to total absorption (>50% even at shortest wavelength), the spectral dependence of absorption in bulk has not shown apparent diurnal variation."

L269-272

*3. Line 248- is r>0.4 considered as a high correlation?*

**Reply:** We use $r$=0.4 to distinguish the factors with relatively higher correction with BrC absorption within the five factors. It is changed as:

"MLR on the total BrC shows relatively higher correction ($r$>0.4) with the factors of HOA, BBOA and OOA2, suggesting the potential importance of the primary biomass burning and traffic source along with OOA2 in governing absorption of BrC."

L251

*4. Line 226 - Do the MAC values of BC match well with those in literature?*

**Reply:** We have added related discussions.

"MAC of BC at $\lambda$=375nm showed to be at 8.4 -16.6 $m^2\,g^{-1}$ with enhanced absorption when high coatings, which was consistent with previous studies which reported $MAC_{BC}$ of 8-10 $m^2\,g^{-1}$, and higher value of 9.7 -17.2 $m^2\,g^{-1}$ under polluted condition (Ding et al., 2019; Hu et al., 2021)."

L222-223

*5. Lines 277-278 – The authors use the comparison of diurnal profiles, the reduction in the absorbance and the absorption coefficient per unit POA mass, as strong evidence of photobleaching of primary BrC. While photobleaching maybe one of the reasons for the observed reduction, the given evidence is not strong enough to make this an absolute conclusion. It is important to discuss if there are other possibilities for this. For an example, it possible that some primary species transformed into less absorbing secondary BrC species, and this could be via other reaction pathways. The type of HOA/BBOA contributing to the absorption during this period may have lower absorptivity. While it may be difficult to provide solid evidence for all the claims, it is important to discuss the possible reasons for such observations.*

**Reply:** We thank reviewer to point this out. The suggestion about the conversion of primary OA to secondary OA with weaker absorption can indeed support our conclusion. We have added the suggested related discussions in the revision.

"In addition to photobleaching, it possible that some primary species transformed into less

absorbing secondary BrC species. During this period, the type of HOA or BBOA that contribute to absorption may also have a lower absorptivity."

L288-289

*Also, considering that BBOA is more absorbing per unit mass than traffic-related OA and has a corresponding peak at night, it is surprising to see the low, almost constant levels of abs/POA from 6-9pm. I believe the reasons for this have not been discussed in the manuscript.*

**Reply:** We have added related discussions in the revision.

"Both HOA and BBOA had night peaks at 6-9pm with HOA having a higher concentration than BBOA. The HOA/BBOA ratio almost unvaried in the diurnal pattern, thus had not resulted in a significant variation of $\sigma_{abs,priBrC}$/POA (Fig. 1m, Fig. 1o and Fig. 4b.). "

L291-293

*Furthermore, while the method used to apportion primary and secondary BrC has been previously used, it may be important to point out that it may not be as straightforward to separate primary and secondary sources of BrC by assuming that all primary BrC are from combustion sources and that there is no cooccurrence of primary and secondary BrC from these. In fact, authors themselves mention that their study and those previously do observe a cooccurrence of these emissions (Lines 272-273). Therefore, it is possible that some of the primary sources are being attributed to secondary sources and vice versa. This maybe a possible reason for the simultaneous peak observed for primary and secondary BrC during morning rush hour.*

**Reply:** We agree with the reviewer that it is possible that some primary sources are attributed to secondary sources or vice versa, which may explain the simultaneous peak observed for primary and secondary BrC during morning rush hour. Related discussions are added.

"The morning peak coinciding with the primary BrC may result from the rapid formation of BrC from sources when emitted gases condensed and formed aerosols. These may lead to high cooccurrence between primary and secondary BrC. Previous studies in urban environment also observed concurrent peaks of primary and secondary BrC, which usually occurred at morning rush hour (Zhang et al., 2020). Furthermore, it is possible that some primary sources are attributed to secondary sources. This may explain the simultaneous peak observed for primary and secondary BrC during the morning rush hour."

L278-284

*6. I am unable to follow how the authors determined that there is a 20% decrease in primary and 30% increase in secondary BrC absorbance due to photooxidation. If I understand correctly, it says in the methodology that this is compared to the overall average absorbance of primary and secondary BrC absorbance. If this is the case, how is it determined that photooxidation alone was responsible for the increase or decrease when there are various*

*processes taking place throughout the day that affect BrC absorbance. I notice that previous reviewers have also raised this concern, but I don't believe it has been properly addressed in the revised manuscript.*

**Reply:** We agree with reviewer that there are multiple processes in the daytime such as photooxidation, photolysis or other photochemical processes may play roles on modifying the absorbance of brown carbon. We have therefore changed the term photooxidation as multiple photochemical processes for these discussions. In addition, we have more clearly demonstrated how the values are obtained. Related discussions are revised.

"The ==photochemical processes== were found to result in reduced contribution of ==fraction of total absorbance of== primary BrC about 20% but enhanced contribution of secondary BrC by 30%."

"==Fig 4e-f shows the photochemical processes led to an enhanced contribution of secondary BrC to the total absorption by 30% from the morning rush-hour to midday, but during the same time reduced the contribution of primary BrC to the total absorption about 20%.=="

"Overall, by apportioning the absorption of primary and secondary BrC, we found the ==photochemical processes== led to an enhanced contribution of ==fraction of total absorbance of== secondary BrC by 30% but reduced contribution of primary BrC about 20% in the semi-urban environment."

L22, L298-301, L315-316

*7. Figure 4 (d-e) –Authors may consider to use "Fraction of total absorbance" or a similar title that better describes what the axis represents*

*This is the case throughout the text where, when a fraction or a percentage is used. Ideally, the description should include the denominator. It may be particularly important to describe it properly when the percentage decrease and increase of primary and secondary BrC absorbance is used to describe the effect of photochemical reactions (Lines 22-23 and 300-301) as this is one of the primary claims.*

**Reply:** We have revised the title of Figure 4 (d-e) and have revised related discussions.

"The ==photochemical processes== were found to result in reduced contribution of ==fraction of total absorbance of== primary BrC about 20% but enhanced contribution of secondary BrC by 30%, implying the concurrent whitening and darkening of BrC."

"Overall, by apportioning the absorption of primary and secondary BrC, we found the ==photochemical processes== led to an enhanced contribution of ==fraction of total absorbance of== secondary BrC by 30% but reduced contribution of primary BrC about 20% in the semi-urban environment."

L22, L315-316

[Figure]

**Figure 4. Diurnal variations of absorption coefficient at λ=375nm ($\sigma_{abs,375}$) for BC (a), primary BrC and absorption efficiency of primary BrC ($\sigma_{abs,priBrC}$)/POA is shown in shade (b), and secondary BrC, along with the $C_xH_yN_z$ and $C_xH_yN_zO_p$ fragments (c); the respective fraction in total for the segregated $\sigma_{abs,375}$ (d-f), with direct radiation shown in shade. In each plot, the lines, dots and whiskers denote the median, mean and the 25th/75th percentiles at each hour respectively.**

*Minor comments :*

*(i) The sentences in the abstract are too long and the message is confusing. Please rephrase. For example;*

*" The absorption of BC is constrained by its size distribution and mixing state, being subtracted from total absorption to obtain the absorption of BrC, then by applying the least-correlation of BC absorption with secondary BrC, the absorption contributed by BC, primary BrC and secondary BrC was apportioned" can be rephrased as;*

*"The absorption of BC is constrained by its size distribution and mixing state and the BrC absorption is obtained by subtracting the BC absorption from the total aerosol absorption. Aerosol absorption was further apportioned to BC, primary BrC and secondary BrC by applying the least-correlation between secondary BrC and BC."*

*Several other instances where the messaging is unclear due to long sentences can be found throughout the manuscript. Please try to write in clear and concise sentences to get the message across more clearly.*

**Reply:** This is revised.

==“The absorption of BC is constrained by its size distribution and mixing state and the BrC absorption is obtained by subtracting the BC absorption from the total aerosol absorption. Aerosol absorption was further apportioned to BC, primary BrC and secondary BrC by applying==

the least-correlation between secondary BrC and BC."

"These primary BrC has a range of absorptivity, which was found to be controlled by burning phases. OA co-emitting with BC (the flaming phase) exhibited a higher absorptivity than OA-dominated smoldering phase (Liu et al., 2021)."

L16-18, L35-36

*(ii) Several grammar, spelling mistakes and missing words can be found. Please read thoroughly to minimize the errors. I am only listing a few examples.*

*e.g.: Line 27 - Atmospheric absorbing organic aerosol (OA), known as brown carbon (BrC), is "a" important contributor to anthropogenic 28 absorption besides black carbon (BC)*

*Line 34 – These primary BrC "has" ....(again long sentence here and the message is unclear).*

*Line 46- "Existing" chromophores*

**Reply:** This is revised.

L27, L34, L46

*(iii) The word brown carbon used in the middle of text after acronym BrC is first introduced. Please be consistent.*

**Reply:** This is revised.

L38

*(iv) Line 37 - Dasari, Sanjeev, et al. "Photochemical degradation affects the light absorption of water-soluble brown carbon in the South Asian outflow." Science advances 5.1 (2019) also discuss photochemical degradation of South Asian outflow.*

**Reply:** This is added.

L37

*(v) Line 39 - "decease" should be changed to "decrease" ....and photobleaching "of"*

**Reply:** This is revised.

L39

**References**

Wang, Q., Han, Y., Ye, J., Liu, S., Pongpiachan, S., Zhang, N., Han, Y., Tian, J., Wu, C., Long, X., Zhang, Q., Zhang, W., Zhao, Z., and Cao, J.: High Contribution of Secondary Brown Carbon to Aerosol Light Absorption in the Southeastern Margin of Tibetan Plateau, Geophysical Research Letters, 46, 4962-4970, 10.1029/2019gl082731, 2019.

---

## Author Response (AR3)

Dear Editor and Reviewer,

We are thankful for the comprehensive comments from Reviewer and Editor for the manuscript. We have now addressed all of the comments as below. The corresponding changes in the texts are ==highlighted in yellow==.

**Reviewer 3:**

*The manuscript by Li et al. titled "Concurrent photochemical whitening and darkening of brown carbon" describes the behavior of primary and secondary brown carbon (BrC) from field observation in a sub-urban site near Beijing, China. The total aerosol absorption is apportioned between black carbon (BC), primary BrC and secondary BrC. Traffic and biomass burning are identified as main sources of primary BrC and nitrogen-containing oxygenated organic aerosol are identified as the main source of secondary BrC. A reduction is observed in primary BrC absorbance/total absorbance during the day time with a simultaneous increase in secondary BrC absorbance/total absorbance. Their main finding that there is field evidence of concurrent whitening and darkening of BrC due to photochemical processes, is primarily based on this observed diurnal variation.*

*While there is some improvement from the previous version with the inclusion of other possible explanations for the observations and in acknowledging the uncertainties in the used method, it is questionable whether the authors have provided sufficient evidence that justify their main findings are indeed evidence of whitening and darkening of BrC, rather than formation and degradation of BrC.*

*Main Comments:*

*1. In Figure 4b), authors have shown the reduction in absorption coefficient of primary BrC/POA (i.e. the MAC of POA) as evidence for the whitening of primary BrC. While there are many uncertainties and other possibilities for this, it is offering some level of evidence for the "whitening" (a better term may be degradation) of primary BrC. However, the same is not shown for secondary BrC. The term "darkening" indicates that the formed secondary BrC increased their absorbance. The increase in secondary BrC abs/total abs only indicates that the absorbance from secondary BrC increased during this time. Due to the very obvious formation of SOA from photochemical processes will indeed result in this. However, it does not indicate that there was a "darkening" process. What does the diurnal variation of MAC of SOA look like? Are you able to see an increase in MACSOA? As OA factors from PMS and BrC absorbance (and primary and secondary BrC absorbance) is available, is it possible to investigate the MAC of the individual OA factor? Do their diurnal variations offer additional information of the "whitening and darkening" effect.*

**Reply:** We thank reviewer to point this out. According to reviewer's suggestion, we have now also calculated the MAC of SOA, as shown below. We can derive the absorption of primary and secondary OA using the method here, but it is unable to derive the absorption or MAC for each PMF factor (only for total POA and SOA). The enhancement of $MAC_{SOA}$ was found to peak in the afternoon. We agree with reviewer that the enhancement of SOA absorptivity may be from the production of BrC rather than only darkening the exiting BrC, we therefore have

changed the term from darkening to enhancement when necessary in the revision. Fig. 4 is now incorporated with $MAC_{SOA}$.

[Figure]

**Diurnal variation of MAC for POA and SOA.**

The related discussions are revised: "Fig. 4b showed that the MAC of POA decreased after the morning peak. The MAC of SOA showed an afternoon peak (Fig. 4c), indicating the enhancement of absorption efficiency of secondary BrC, which occurred in a few hours after the peak solar radiation."

L321-323

*2. Lines 269-271-While the overall AAE of total absorbance may be highly influenced by BC absorbance, the AAE of total BrC, primary BrC and secondary BrC can be individually calculated as the total absorbance has already been apportioned.*

**Reply:** The mean AAE during the experiment is now obtained through a power fitting on the average absorption coefficient at different wavelengths for total BrC, primary BrC and secondary BrC. We have added related discussions.

"The mean AAE of total BrC, primary BrC and secondary BrC is obtained by power fitting on the mean absorption coefficient during the experiment (Fig.S7), which is 6.2, 5.7 and 6.4 respectively. This is consistent with other studies that SOA usually had a higher AAE than POA (Gilardoni et al., 2016; Jiang et al., 2022)."

[Figure]

**Figure S7. Absorption coefficient at λ=375, 470, 528 and 635 nm for total BrC, primary BrC and secondary BrC. The AAE is obtained by power fitting on the spectra of absorption coefficient.**

L270-272

*3. Line 291-293 – The meaning is very unclear.*

**Reply:** This sentence is now removed.

*4. Line 22 and line 316- "fraction of total absorbance of secondary BrC" is unclear. Do you mean fraction of absorbance of secondary BrC to total absorbance?*

**Reply:** We have now clarified this point according to reviewer's suggestion.

"The photochemical processes were found to result in reduced contribution of fraction of absorbance of primary BrC to total absorbance about 20% but enhanced contribution of secondary BrC by 30%, implying the concurrent whitening and darkening of BrC."

"Overall, by apportioning the absorption of primary and secondary BrC, we found the photochemical processes led to an enhanced contribution of fraction of absorbance of secondary BrC to total absorbance by 30% but reduced contribution of primary BrC about 20% in the semi-urban environment."

L22-23, L320

*5. Lines 327-336 - As the primary focus of the paper is the "whitening and darkening" of BrC, a more detailed analysis with further evidence is required to make a strong conclusion on this. The 4. Conclusions section does not even include this main point and does not offer any atmospheric implication of the major observations from this manuscript.*

**Reply:** The MAC of secondary BrC is now added to aid the conclusion. The conclusion is also revised to indicate the atmospheric implication about the concurrent bleaching and formation

of BrC.

"This study apportioned the shortwave absorption of BC, primary and secondary BrC, through concurrent measurements of BC microphysical properties and OA mass spectra. The apportioned primary BrC absorption was linked with traffic and biomass burning emissions, while secondary BrC was found to be associated with an oxygenated secondary OA factor with higher nitrogen content. The enhancement of secondary BrC and decease of primary BrC simultaneously occurred via daytime photooxidation. The results emphasize the importance of nitrogen-containing OA in contributing to BrC. These OA could primarily emit as aerosol phase, or in gas phase which requires further oxidation to be in aerosol phase to serve as BrC. The $NO_x$-involved chemistry is prone to add nitrogen element to the existing OA and enhance the absorptivity of chromophores. The anthropogenic $NO_x$ emission could be therefore an important source in producing shortwave absorbing components in the atmosphere, which may offset some of the conventionally-thought photobleaching of BrC by photochemistry. The production of secondary BrC should be considered when assessing the environment and climate impacts of light-absorbing aerosols."

L338-339, L342-344

**Editor:**

*There are still a couple major concerns from the reviewer regarding the results interpretation that requires further improvement/clarification. In addition, I would like to include a few additional major comments as listed below.*

*1) Further justification is required for the use of a single value of Sigma(abs,pri) (i.e. determined by equation 4) as two primary BrC sources (i.e., traffic and biomass burning) are identified in this work. The OA from these two primary emissions likely have different BrC absorption and characteristics. Although the argument is provided (lines 291-293) to claim that the HOA/BBOA ratio almost unvaried in the diurnal pattern, Figure 1 clearly shows that the HOA/BBOA ratios are different between the morning rush hour and the night-time peak. This is very important to provide stronger justification here as the subsequent calculation can have significant impact on the arguments for photobleaching of primary BrC and formation of secondary BrC associated with the primary emissions in page 12.*

**Reply:** We thank editor to point this out. We agree with editor that there may be different primary BrC/BC ratio between HOA and BBOA sources and this may lead to bias in deriving the subsequent results. We have more carefully investigated the diurnal pattern of HOA and BBOA, and found only a slight morning rush-hour peak for HOA (though bearing considerable variation). A further investigation on the HOA/BBOA ratio found no apparent diurnal pattern (bearing large variation), shown as below. The source difference is therefore not considered to have significantly influenced the diurnal pattern of derived parameters. In addition, this method is only valid with sufficient data points thus we may only obtain a single mean value for the entire experiment, which represents the mean $BrC_{pri}/BC$ in this environment during the experimental period.

[Figure]

**Diurnal variation of HOA, BBOA and HOA/BBOA.**

Other literatures using this method also derived the mean value of $BrC_{pri}/BC$ for the urban environment influenced by multiple sources including traffic, coal combustion and biomass burning (Wang et al., 2019; Wang et al., 2020; Gao et al., 2022).

*2) Lines 285-287: More detail interpretation is required for assigning photobleaching as a key aging process of primary BrC peak observed at nighttime. Why the key aging process involved*

*has to be photobleaching? The logic flow of the discussion implies that the changes in primary BrC absorption occurred locally or in short time scale. If so, please provide more detail clarification. Can the observation cause by regional transport of aged BBOA as well for example? What are the implications in terms of BrC chemistry/characteristics if the BrC absorptivity deceased faster than the HOA and BBOA mass?*

**Reply:** We thank editor to point out these important points. More clarification is now added to explain the decreased absorptivity of primary BrC. In the daytime, the BrC may react with OH radical in aerosol phase, or by enhanced evaporation and reaction in gas phase, and maybe further decreased by aqueous reaction when higher RH at night. All these may contribute to the decrease of chromophores. We agree with editor that some more aged BBOA source could be mixed with fresher HOA source, though there may be also some aged HOA. The aging scale of these sources is unable to be resolved in this study, but this is not likely to affect the diurnal pattern (which tends to be more influenced by local sources) we investigated, thus not affecting the conclusion.

More discussions about the implications of BrC absorptivity decrease are also added.

"The night had contributions from BC and primary BrC at 50±2% and 30±3% respectively, with 20±3% as secondary BrC. Fig. 4b showed the decrease of primary BrC absorption tended to be more rapid than the HOA and BBOA mass (even a slight increase for HOA Fig. 1m and Fig. 1o) in the midday, leading to decreased absorption coefficient per unit mass of primary BrC (shade in Fig. 4b), which indicates the decrease of BrC absorptivity likely due to photochemistry. This may involve the OH radical reaction with existing chromophores in aerosol phase (Schnitzler et al., 2020) or by enhanced evaporation of aerosols to gas phase (Palm et al., 2020) leading to further decrease of BrC absorptivity during midday. In addition to photobleaching, it possible that some primary species transformed into less absorbing secondary BrC species. During this period, the type of HOA or BBOA that contribute to absorption may also have a lower absorptivity. In this context, a recent chamber study reported that the primary BrC from biomass burning plumes could be bleached to half of the initial absorptivity in 2-3 hours (Liu et al., 2021). The reaction of BrC with OH radical has been widely recognized as the main pathway for the loss of primary BrC absorptivity (Liu et al., 2020), and was parameterized as an exponential decrease with time at certain OH radical concentration in global scale (Wang et al., 2018b)."

L288-291, L295-297

*3) Lines 315-321: As the key message of this work is concurrent photochemical whitening and darkening (or degradation and formation) of ambient BrC as reflected in the title of manuscript. However, the changes in the percentage contribution of primary and secondary BrC to the total absorbance would not be able to support this conclusion. Instead, absolute changes in BrC mass and/or absorbance (e.g. MAC of total OA or PMF factors) are likely required to support this conclusion and they should be clearly presented in the manuscript.*

**Reply:** We thank editor to point this out. In addition to the changes in the percentage contribution of primary and secondary BrC to the total absorbance, we have also added the MAC (absorption per unit mass of OA) of POA and SOA to imply the change of their

absorptivity. We can derive the absorption of primary and secondary OA using the method here, but it is unable to derive the absorption for each PMF factor (only for total POA and SOA). The enhancement of $MAC_{SOA}$ was found to peak in the afternoon. Fig. 4 is now incorporated with $MAC_{SOA}$.

[Figure]

**Diurnal variation for MAC of POA and SOA.**

These are now added in the revision: "Fig. 4b showed that the MAC of POA decreases after the morning peak. The MAC of SOA showed an afternoon peak (Fig. 4c), indicating the enhancement of absorption efficiency of secondary BrC, which occurred in a few hours after the peak solar radiation."

L321-323

**References**

Gao, Y., Wang, Q., Li, L., Dai, W., Yu, J., Ding, L., Li, J., Xin, B., Ran, W., Han, Y., and Cao, J.: Optical properties of mountain primary and secondary brown carbon aerosols in summertime, Science of the Total Environment, 806, 10.1016/j.scitotenv.2021.150570, 2022.

Wang, Q., Liu, H., Wang, P., Dai, W., Zhang, T., Zhao, Y., Tian, J., Zhang, W., Han, Y., and Cao, J.: Optical source apportionment and radiative effect of light-absorbing carbonaceous aerosols in a tropical marine monsoon climate zone: the importance of ship emissions, Atmos Chem Phys, 20, 15537-15549, 10.5194/acp-20-15537-2020, 2020.

Wang, Q., Ye, J., Wang, Y., Zhang, T., Ran, W., Wu, Y., Tian, J., Li, L., Zhou, Y., Ho, S. S. H., Dang, B., Zhang, Q., Zhang, R., Chen, Y., Zhu, C., and Cao, J.: Wintertime Optical Properties of Primary and Secondary Brown Carbon at a Regional Site in the North China Plain, Environmental Science & Technology, 53, 12389-12397, 10.1021/acs.est.9b03406, 2019.

---

## Author Response (AR4)

Dear Editor and Reviewer,

We are thankful for the comprehensive comments from Reviewer and Editor for the manuscript. We have now addressed all of the comments as below. The corresponding changes in the texts are highlighted in yellow.

**Reviewer 3:**

*The manuscript titled "Concurrent photochemical whitening and darkening of brown carbon" by Li et al. describes the behavior of primary and secondary brown carbon (BrC) from field observation in a sub-urban site near Beijing, China. The main finding of this work is that it is offering field evidence of concurrent whitening and darkening of ambient BrC.*

*The reviewers and editor have raised several concerns and have given suggestions for improvement, most of which have been addressed by the authors. However, it appears that the authors' have not fully grasped the reasoning for these comments and have failed to modify the manuscript in a significant way. Some of the comments have been addressed by adding one or two sentences in the suggested paragraph and the changes are not reflected anywhere else in the manuscript, even if it changes some of the main messaging. Take for example the comment by editor that additional evidence to justify the main claim of concurrent whitening and darkening, other than the percentage difference in primary and secondary BrC absorption, is needed. This has only been addressed in one line in the accompanying text, suggesting the enhancement in MAC of SOA, but not addressed in the abstract, where it only describes the percentage difference.*

**Reply:** We have now added more discussions in the abstract and expanded discussions in the main texts.

In the abstract:

[revised manuscript text omitted]

L340-343

*Furthermore, comments have been made regarding calculating the MAC of individual OA factors, and has not been addressed. With source-apportioned OA factors and total OA absorbance, several studies have used methods such as multiple linear regression analysis (Qin et al. "Chemical characteristics of brown carbon in atmospheric particles at a suburban site near Guangzhou, China." ACP 18.22 (2018): 16409-16418. Wang et al. "Wintertime optical properties of primary and secondary brown carbon at a regional site in the North China Plain." ES&T 53.21 (2019): 12389-12397. Kasthuriarachchi et al. "Light absorbing properties of primary and secondary brown carbon in a tropical urban environment." ES&T 54.17 (2020): 10808-10819. Wang et al. "Aqueous production of secondary organic aerosol from fossil-fuel emissions in winter Beijing haze." PNAS 118.8 (2021): e2022179118 and latest studies with ridge regression models (Zhang et al. "Impact of COVID-19 lockdown on the optical properties and radiative effects of urban brown carbon aerosol." Geo. Front. 13.6 (2022): 101320.) to obtain MAC of individual OA factors. The fact that authors have failed to do the same is questionable. The addition of this information has the potential to offer supporting evidence to the main claim of this work as it will help to identify how the MAC of each OA factor changed during the day.*

*A complete revision of this work, including MAC of individual factors and addressing the uncertainties of the calculation, assumptions etc., with more focus on the light absorbing properties in the Results and Discussion section (more than half is attributed to OA source discussion), with a thorough language revision may be re-submitted for revision.*

**Reply:** We thank reviewer to point this out. By rechecking the manuscript, we found we have actually performed the multiple linear regression in the previous version, as reviewer suggested. We have now reemphasized this point and added related discussions.

"A multiple linear regression (MLR) analysis is performed to apportion the absorption coefficient of BrC with the PMF attributed OA factors, expressed as:

$$\sigma_{abs,BrC} = a_0 + a_1 \cdot [OOA1] + a_2 \cdot [OOA2] + a_3 \cdot [BBOA] + a_4 \cdot [COA] + a_5 \cdot [HOA] \tag{6}$$

where $a_1$ to $a_5$ represents the regression coefficients for each factor. These coefficients can be associated with the absorptivity of each factor, i.e., a larger coefficient implies a higher MAC for the source associated with that OA factor (Kasthuriarachchi et al., 2020; Wang et al., 2021). The BBOA was found to have the highest MAC at 2.59 $m^2\ g^{-1}$, consistent with previous studies which also found significantly higher absorption for biomass burning source (Qin et al., 2018; Wang et al., 2019b; Zhang et al., 2022). The other POA factors generally have a higher MAC than SOA (the MAC of HOA and COA are is 1.70 $m^2\ g^{-1}$ and 1.30 $m^2\ g^{-1}$, respectively). Particularly, the OOA2 has a relatively high MAC of 1.22 $m^2\ g^{-1}$, which is likely to result from the production of secondary BrC as discussed below. The contribution of each source-specific OA factor to $\sigma_{abs,BrC}$ can also be obtained. This analysis is performed for the total BrC, primary and secondary BrC respectively. The results are shown in Table 1."

L258-265

**Editor:**

*Thanks for your detail responses to my comments for your last version, and I think you have addressed my questions appropriately. For the next revision, I suggest to add the diurnal variation of HOA/BBOA and related justification in SI so that reader can closely follow the detail of your data analysis approach. The updated Figure 4 certainly enhance the supporting argument for the main message of this work.*

**Reply:** We are thankful to Editor for the comprehensive comments to our manuscript. The diurnal variation of HOA/BBOA is now added in Figure S8.

*However, the reviewer#3 still have some major concerns in terms of presentation quality and data analysis as outlined in the reviewer's report. Although the reviewer recommends to reject this work, I think this work can be potentially important to advance our understanding of BrC sources and formation in the atmosphere if the reviewer's comment can be addressed appropriately.*

We thank Editor for the opportunity to revise our manuscript, and we believe in this version we will fully address all of the issues reviewer#3 raised.

*In my point of view, there are still two major actions/justification requested by the reviewer#3 to enhance the quality of this manuscript.*

*1) The reviewer requests to check all the scientific arguments are consistently presented throughout the manuscript as some of the conclusion and details might be changed along the review process.*

**Reply:** We have now very carefully revisited the manuscript once again to make sure all the related discussions have been adapted for the updated figures and data analysis. The revisions are listed as below:

[revised manuscript text omitted]

L340-343

*2) The reviewer requests to obtain MAC values for individual OA factor. The authors response to my previous comment that it is unable to derive the MAC values for each OA factor, and I suggest the authors to provide a more clear explanation on this. Nevertheless, my understanding is that the results presented in Table 1 were actually obtained by MLR analysis between total BrC absorption and PMF-OA factors, which is similar to the approach used by other studies (see the reference list from the reviewer). The regression coefficient is likely relevant to the MAC values, which at least can be used to compare the light absorption properties of HOA, BBOA and OOA-2 observed in this study (e.g., BBOA > HOA ~OOA-2), and perhaps those derived in other previous studies.*

**Reply:** We thank Editor to point this out. By rechecking the manuscript, we found we have actually performed the multiple linear regression in the previous version, as Editor suggested. We have now reemphasized this point and added related discussions.

"A multiple linear regression (MLR) analysis is performed to apportion the absorption coefficient of BrC with the PMF attributed OA factors, expressed as:

$$\sigma_{abs,BrC} = a_0 + a_1 \cdot [OOA1] + a_2 \cdot [OOA2] + a_3 \cdot [BBOA] + a_4 \cdot [COA] + a_5 \cdot [HOA] \qquad (6)$$

where $a_1$ to $a_5$ represents the regression coefficients for each factor. These coefficients can be associated with the absorptivity of each factor, i.e., a larger coefficient implies a higher MAC for the source associated with that OA factor (Kasthuriarachchi et al., 2020; Wang et al., 2021). The BBOA was found to have the highest MAC at 2.59 m$^2$ g$^{-1}$, consistent with previous studies which also found significantly higher absorption for biomass burning source (Qin et al., 2018; Wang et al., 2019b; Zhang et al., 2022). The other POA factors generally have a higher MAC than SOA (the MAC of HOA and COA are is 1.70 m$^2$ g$^{-1}$ and 1.30 m$^2$ g$^{-1}$, respectively). Particularly, the OOA2 has a relatively high MAC of 1.22 m$^2$ g$^{-1}$, which is likely to result from the production of secondary BrC as discussed below. The contribution of each source-specific OA factor to $\sigma_{abs,BrC}$ can also be obtained. This analysis is performed for the total BrC, primary and secondary BrC respectively. The results are shown in Table 1."

L258-265

---

## Author Response (AR5)

Dear Editor and Reviewers,

We are thankful for the comprehensive comments from Reviewer and Editor for the manuscript. We have now addressed all comments and revised our previous manuscript accordingly. The corresponding changes in the texts are ==highlighted in yellow==.

**Reviewer 3:**

*Referee report on ACP-2022-483 - Concurrent photochemical whitening and darkening of ambient brown carbon*

*Appreciate the effort of the authors to address the comments in detail and incorporate them in the necessary content throughout the manuscript. I believe the manuscript is suitable for publication after the following minor comments are addressed. Overall, as there are several instances of long sentences and grammatical errors that cloud the important messaging, a thorough language revision may be considered.*

We thank reviewer for the generally positive comments for the current version of the manuscript. We have now addressed all the comments raised.

*Minor Comments:*

*1. Lines 22-25-MAC and contribution to total absorbance of POA and SOA could be discussed for POA and SOA separately to make the message clearer. For example. "Photochemical processes were found to reduce the mass absorption cross section (MAC) of primary OA, reducing its contribution to total absorption by 20%, at the same time increasing MAC for secondary OA, which showed a 30% enhancement in contribution to total absorbance….."*

**Reply:** This is revised.

==“The photochemical processes were found to reduce the mass absorption cross section (MAC) of primary OA, reducing its contribution to total absorption by 20%, at the same time increasing MAC for secondary OA, which showed a 30% enhancement in contribution to total absorbance, implying the concurrent whitening and darkening of BrC.”==

L23-25

*2. Line 25-26-"The study provides….…..nitrogen-containing secondary OA can compensate for some effects of bleaching of primary BrC."*

*In some text, the authors have implied that photooxidation was responsible for the observed changes while in other instances photochemical processes are mentioned. As the study cannot isolate what type of reactions may have caused the increase or decrease in absorptivity, maybe it is safer to use the term photochemical processes. Please read through and keep the message consistent.*

**Reply:** We agree with reviewer that the study is unable to isolate what type of reactions may have caused the increase or decrease in absorptivity. We have therefore changed the term

photooxidation as photochemical processes. Related discussions are revised.

"The above findings mean the enhancement or bleaching of BrC absorptivity via ==photochemical processes== will coexist."

"The slight enhancement at noon for OOA1 (also for OOA2) soon after morning rush-hour indicated the likely rapid formation of SOA through ==photochemical processes==."

"This aging or oxidation likely occurred through ==photochemical processes== during early afternoon and aqueous processes (high RH conditions) during nighttime."

"The high NOx emission of traffic VOCs may have largely involved nitrogen chemistry ==in the photochemical processes==."

"The daytime formation of organic nitrate may follow the gas-phase ==photochemical processes== in which the excess NO could add to the peroxy radical to produce organic nitrate (Liebmann et al., 2019)."

"We found the ==photochemical processes== decreased the MAC of POA but increased the MAC of SOA, resulting in an enhanced contribution of secondary BrC to total absorbance by 30% but reduced contribution of primary BrC about 20% in the semi-urban environment. This revealed that the whitening and darkening of BrC occurred simultaneously, and the secondary BrC produced by ==photochemical processes== may compensate some bleaching effect of primary BrC."

L50, L214, L332, L335, L337, L354, L357

*3. Line 318-321 -The sentences are not well combined. Re-writing these sentences combining the messages maybe clearer.*

*4. Line 321- Needs revision for grammatical errors and plain language.*

**Reply:** This is revised.

"==Fig. 4b-c showed that the MAC of POA decreased after the morning peak, but the MAC of SOA had an afternoon peak. This indicated the enhancement of absorptivity of secondary BrC, which occurred in a few hours after the peak solar radiation. These results implied the photochemical processes decreased the absorptivity of POA but increase for of SOA.=="

L322-324

*5. Line 324- Can be considered to be revised as "Though other processes such as aqueous-phase reactions may cause changes to MAC of BrC at nighttime, the apparent change in aerosol absorption observed in this study during daytime can play an important role on the radiative impacts due to intensive solar radiation during daytime"?*

**Reply:** Thanks for the suggestion. This is revised.

"==Though other processes such as aqueous-phase reactions may cause changes to the MAC of BrC at nighttime, the apparent change in aerosol absorption observed in this study can play an important role on the radiative impacts due to intensive solar radiation during daytime.=="

*6. Lines 314-339 – If possible, it may be better to revise this paragraph and re-organise the important information mentioned. The current layout appears to be jumping back and forth between MAC, relative contribution of POA and SOA and SOA formation pathways.*

**Reply:** This paragraph is now reorganized.

"The night and morning peak of OOA2 and the morning peak of the absorption coefficient of secondary BrC ($\sigma_{abs,secBrC}$) may result from primarily emitted moderately oxygenated OA, which was reported from some diesel sources (Dewitt et al., 2015; Gentner et al., 2012). Besides the morning rush-hour peak, there was an early afternoon peak for $\sigma_{abs,secBrC}$ prevailing the dilution effect of daytime boundary layer (Fig. 4c-S5). The fraction of $\sigma_{abs,secBrC}$ in total $\sigma_{abs}$ thus had a pronounced early afternoon peak soon after the peak solar radiation (Fig. 4f), and a peak after midnight soon after the nighttime peak of primary BrC (Fig. 4e).
Fig. 4b-c showed that the MAC of POA decreased after the morning peak, but the MAC of SOA had an afternoon peak. This indicated the enhancement of absorptivity of secondary BrC, which occurred in a few hours after the peak solar radiation. These results implied the photochemical processes decreased the absorptivity of POA but increase for of SOA. Fig 4e-f showed the photochemical processes led to an enhanced contribution of secondary BrC to the total absorption by 30% from the morning rush-hour to midday, but during the same time reduced the contribution of primary BrC to the total absorption about 20%. Though other processes such as aqueous-phase reactions may cause changes to the MAC of BrC at nighttime, the apparent change in aerosol absorption observed in this study can play an important role on the radiative impacts due to intensive solar radiation during daytime.
Table 1 showed the SOA compounds containing nitrogen (i.e., the OOA2) considerably contributed to the light absorption. The shift of peaking time from primary to secondary BrC demonstrates the possible processes of SOA formation, such as from gases. This aging or oxidation likely occurred through photochemical processes during early afternoon and aqueous processes (high RH conditions) during nighttime. The oxidized volatile organic compounds (VOCs) with nitrogen chemistry involved could condense to produce additional mass in particle phase (Ehn et al., 2014; Finewax et al., 2018). The high $NO_x$ emission of traffic VOCs may have largely involved nitrogen chemistry in the photochemical processes. Previous studies found the $NO_x$-involved SOA could produce considerable chromophores containing nitro-aromatics in hours (Wang et al., 2019d; Keyte et al., 2016). The daytime formation of organic nitrate may follow the gas-phase photochemical processes in which the excess NO could add to the peroxy radical to produce organic nitrate (Liebmann et al., 2019). The nighttime chemistry involves $NO_3$ radical through the oxidation of $NO_2$ by $O_3$, through which the organic nitrate could be produced by initializing the production of nitrooxy peroxy radicals (Ng et al., 2008; Rollins et al., 2012). Laboratory studies also widely observed the rapid production of nitrogen-containing OA through $NO_x$ chemistry, which could contribute to light absorption of aerosols (Nakayama et al., 2013; Liu et al., 2015c)."
L316-342

*7. Line 340-342 – Sentence too long and unclear. Please re-write as two or more sentences.*

*8. Line 340-348 – Can be considered to be included in Conclusions section instead, as it is repeating some of the main messages that was described in the paragraph right above.*

**Reply:** This paragraph is now moved to the conclusion section.

"This study apportioned the shortwave absorption of BC, primary and secondary BrC, through concurrent measurements of BC microphysical properties and OA mass spectra. The apportioned primary BrC absorption was linked with traffic and biomass burning emissions. Primary OA generally had a higher MAC than secondary OA. OA from biomass burning was found to have the highest MAC in POA factors. Secondary BrC was found to be associated with an oxygenated secondary OA factor with higher nitrogen content. We found the photochemical processes decreased the MAC of POA but increased the MAC of SOA, resulting in an enhanced contribution of secondary BrC to total absorbance by 30% but reduced contribution of primary BrC about 20% in the semi-urban environment. This revealed that the whitening and darkening of BrC occurred simultaneously, and the secondary BrC produced by photochemical processes may compensate some bleaching effect of primary BrC. The dominance of both competing processes may depend on the timescale and altitude in the atmosphere. For example, the enhanced BrC fraction observed above the planetary boundary layer may be explained by the enhanced secondary BrC (Tian et al., 2020), while further aging may bleach the produced chromophores of these SOA.

The results emphasize the importance of nitrogen-containing OA in contributing to BrC. The $NO_x$-involved chemistry is prone to add nitrogen element to the existing OA and enhance the absorptivity of chromophores. The anthropogenic $NO_x$ emission could be therefore an important source in producing shortwave absorbing components in the atmosphere, which may offset some of the conventionally-thought photobleaching of BrC by photochemistry. The production of secondary BrC should be considered when assessing the environment and climate impacts of light-absorbing aerosols."
L350-365

*9. Lines 360-361- "These OA could primarily emit as aerosol phase, or in gas phase which requires further oxidation to be in aerosol phase to serve as BrC." - I am unclear of the relevance of this sentence.*

**Reply:** This is removed.